# DeCoP: Enhancing Self-Supervised Time Series Representation with Dependency Controlled Pre-training

## Abstract

Modeling dynamic temporal dependencies is a critical challenge in time series pre-training, which evolve due to distribution shifts and multi-scale patterns. This temporal variability severely impairs the generalization of pre-trained models to downstream tasks. Existing frameworks fail to capture the complex interactions of short- and long-term dependencies, making them susceptible to spurious correlations that degrade generalization. To address these limitations, we propose **DeCoP**, a **De**pendency **Co**ntrolled **P**re-training framework that explicitly models dynamic, multi-scale dependencies by simulating evolving inter-patch dependencies. At the input level, DeCoP introduces Instance-wise Patch Normalization (IPN) to mitigate distributional shifts while preserving the unique characteristics of each patch, creating a robust foundation for representation learning. At the latent level, a hierarchical Dependency Controlled Learning (DCL) strategy explicitly models inter-patch dependencies across multiple temporal scales, with an Instance-level Contrastive Module (ICM) enhances global generalization by learning instance-discriminative representations from time-invariant positive pairs. DeCoP achieves state-of-the-art results on ten datasets with lower computing resources, improving MSE by 3% on ETTh1 over PatchTST using only 37% of the FLOPs. The source code is available at https://anonymous.4open.science/r/DeCop-62A7.

## 1 Introduction

Time series analysis plays a critical role in applications like weather forecasting (Wu et al., 2023b; Liu et al., 2022), fault detection (Deng & Hooi, 2021; Zhang et al., 2022b), and sales prediction (Wu et al., 2023a; Ekambaram et al., 2020). With abundant unlabeled time series data across domains, pre-training approaches for representation learning without extensive annotation are increasingly popular. Recent research has focused on pretrained models to address tasks like forecasting and classification in a general-purpose backbone (Goswami et al., 2024; Jin et al., 2023; Liu et al., 2024; Woo et al., 2024; Rasul et al., 2023).

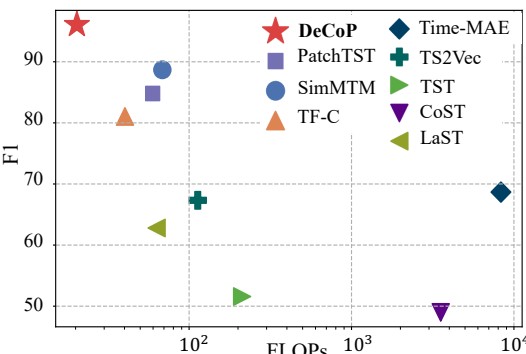

Figure 1: DeCoP consistently outperforms state-of-the-art pretraining frameworks on classification tasks across datasets with lower FLOPs (Floating Point Operations).

Despite these efforts, time series data pose fundamental challenges for self-supervised learning. First, the non-stationary nature of time series induces temporal distribution shifts, causing the underlying dependency patterns between patches to evolve over time. Second, time series inherently exhibit multi-scale temporal structures, encompassing both short-term fluctuations and long-term trends. Models that operate at a single scale consequently fail to capture these rich, hierarchical dependencies. These challenges highlight the need for a framework that can perform

controllable, multi-scale dependency modeling in the presence of distribution shifts, enabling representations that capture both fine-grained local semantics and broader contextual patterns.

However, existing time series pre-training frameworks such as Masked Time series Modeling (MTM) (Nie et al., 2022; Dong et al., 2024; Lee et al., 2023) predominantly rely on single-scale architectures such as Transformers (Vaswani et al., 2017). This architectural choice limits their ability to capture multi-scale temporal dependencies and renders them insensitive to relative positional information. Such limitations can lead to spurious correlations, particularly in long-term dependency modeling, and result in entangled representations that blur local variations and dilute global consistency. Moreover, to mitigate distribution shifts, these approaches often operate at the instance level and apply uniform normalization statistics across all patches, ignoring their distinct local structures (Kim et al., 2021). This coarse-grained normalization results in over-smoothing, suppressing informative temporal patterns such as peaks and abrupt transitions at the patch level.

In this paper, we propose **DeCoP**, a **De**pendency **Co**ntrolled **P**re-training MTM framework that explicitly models dynamic, multi-scale dependencies across time. This controllable learning framework enhances the generalization of time series pretrained models while requiring significantly lower computational cost (FLOPs) in Figure 1. Specifically, at the *input* level, DeCoP applies Instance-wise Patch Normalization (IPN), which incorporate patch-level and instance-level statistics. This enables the model to preserve local semantic variation while stabilizing distribution shifts across time, establishing a more reliable basis for subsequent dependency modeling.

At the *latent representation* level, DeCoP introduces a hierarchical Dependency Controlled Learning (DCL) method to model inter-patch dependencies by dynamically adjusting the temporal receptive field, capturing both short-term and long-range patterns. Concurrently, we introduce an Instance-level Contrastive Module (ICM) that operates on the representations generated by DCL, promoting global alignment for time-invariant positive pairs to improve performance on high-level downstream tasks such as classification. Extensive experiments demonstrate that DeCoP achieves state-of-the-art performance across ten benchmark datasets. The main contributions of this work are as follows:

- We propose **DeCoP**, an efficient dependency controlled pre-training framework that enhances time series representation by explicitly modeling dynamic temporal dependencies under distribution shifts.

- We introduce Instance-wise Patch Normalization, which integrates patch-level statistical information into normalization. This mitigates distributional shifts and preserves local semantic features captured by patch information, providing a stable foundation for modeling dynamic temporal dependencies.

- We develop a hierarchical Dependency Controlled Learning strategy that adaptively captures both short- and long-term dependencies across temporal scales, with a Instance-level Contrastive Module aligning high-level semantic information between time-invariant positive sample pairs, enhancing global semantic learning for high-level downstream tasks.

- DeCoP outperforms existing pretrained models on ten datasets with significantly lower FLOPs, achieving 3% lower MSE than PatchTST on ETTh1 using only 37% of the FLOPs.

## 2 METHOD

**Problem Setting**. Given a univariate time series $x \in \mathbb{R}^L$ with look-back length $L$, the output is masked time series patch $\hat{x}$ and the model is optimized by reconstructing the randomly masked patches by $MSE = \|x - \hat{x}\|_2^2$.

Most existing time series pretraining frameworks follow the transformer structure in Natural Language Processing (Devlin et al., 2019) as shown in Figure 2b. However, this framework overlooks the inherent multi-scale characteristics of time series data and tends to capture spurious dependency correlation due to the dynamical modeling challenge. To overcome this, we present **DeCoP**, a Dependency-Controlled Pre-training framework that improves self-supervised time series representation by modeling dynamic and non-uniform temporal dependencies (Figure 2a). DeCoP integrates modules at both the input and latent levels. At the input level, instance-wise Patch Normalization stabilizes representations by combining local and instance-level statistics. At the latent level, a hierarchical Dependency Controlled Learning method adaptively models inter-patch dependencies

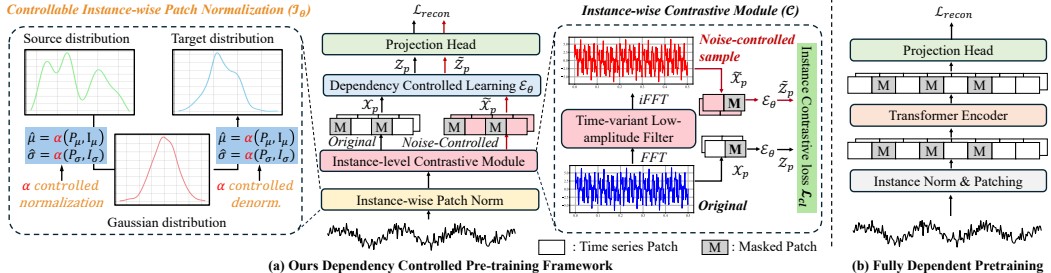

Figure 2: (a) The proposed DeCoP framework addresses dynamic and multi-scale temporal dependencies through a controllable pipeline. IPN incorporates patch-level statistics to stabilize distribution while preserving local semantics. ICM generate time-invariant positive pairs and a contrastive loss to support the hierarchical DCL for multi-scale modeling. DeCoP is optimized with reconstruction loss $\mathcal{L}_{\text{recon}}$ and contrastive loss $\mathcal{L}_{\text{cl}}$. (b) In contrast, traditional frameworks neglect finer-scale statistics and fail to capture the complex interactions of multiscale dependencies.

across multiple temporal scales, and an Instance-level Contrastive module generates time-invariant positive samples to enhance global representation modeling. Together, these components enable robust temporal modeling under varying dependencies.

## 2.1 INSTANCE-WISE PATCH NORMALIZATION FOR STABILIZING INPUT DISTRIBUTIONS

At the input layer, to address the challenges of distribution shift in time series data for time series dependency modeling, we propose Instance-wise Patch Normalization (IPN) in Figure 2a. By integrating patch-level variation with global instance-level distribution information, IPN preserves local semantic features, which are critical for capturing short-term patterns and providing a stable foundation for controlled dependency modeling. Specifically, given a univariate time series $x \in \mathbb{R}^L$, we divide it into patches $\mathcal{X}_p = \{x_1, x_2, ..., x_N\}$ with patch size $P$ and stride $S$. The total number of patches $N$ is given by:

$$N = \left\lfloor \frac{L-P}{S} \right\rfloor + 2. \tag{1}$$

Each patch $x_n \in \mathbb{R}^P$ is an independent unit that captures localized temporal patterns. To quantify the fine-grained variations within each patch, we first compute the patch-wise mean, $E_P[x_n]$, by averaging over its $P$ time points:

$$E_P[x_n] = \frac{1}{P} \sum_{i=1}^{P} x_{n,i}. \tag{2}$$

Subsequently, we compute the patch-wise variance, $Var_P[x_n]$, which measures the dispersion of these points around the mean:

$$Var_P[x_n] = \frac{1}{P} \sum_{i=1}^{P} (x_{n,i} - E_p[x_n])^2. \tag{3}$$

To exploit instance statistics, we leverage distribution information at the instance scale to incorporate global distribution information by calculating mean and variance of $x$:

$$E_I[x] = \frac{1}{L} \sum_{j=1}^{L} x_j, Var_I[x] = \frac{1}{L} \sum_{j=1}^{L} (x_j - E_I[x])^2. \tag{4}$$

where $j$ is the relative index of time series $x$. For each time series, the mean and variance are calculated along the $L$ dimension. After obtaining instance and patch-wise distribution information, a learnable parameter $\alpha \in \mathbb{R}$ is introduced to balance local and global information, controlling their influence as follows:

$$E[x_n] = (1 - \alpha) \times E_I + \alpha \times E_P, \tag{5}$$

Figure 3: (a) The ICM filters time-variant low-amplitude components in the frequency domain to generate denoised positive samples $(\mathcal{X}_p, \tilde{\mathcal{X}}_p)$, which serve as stable semantic patterns to facilitate global dependency modeling. (b) The DCL module performs controlled dependency learning across multiple temporal scales, enabling adaptive modeling of inter-patch dependencies under varying temporal dynamics. Inter-patch dep.: inter-patch dependency.

where $E_I$ and $E_P$ are the global mean and local mean of $x_n$, respectively. The calculation of final variations is given by:

$$Var\left[x_n\right] = (1 - \alpha) \times Var_I + \alpha \times Var_P. \tag{6}$$

Finally, each patch $x_n$ is transformed into $\tilde{x}_n$ using the computed mean and variance for instance-wise patch normalization:

$$\tilde{x}_n = \frac{x_n - E\left[x_n\right]}{\sqrt{Var\left[x_n\right] + \epsilon}}. \tag{7}$$

This results in a normalized sequence $\mathcal{X}_p = \{\tilde{x}_1, \tilde{x}_2, ..., \tilde{x}_n, ..., \tilde{x}_N\}$. During the pretraining stage stage, we reconstruct the normalized time series $\mathcal{X}_p$. During the finetuning stage, we return the mean and variance back through denormalization. This process establishes a stable foundation for dependency modeling, effectively overcoming the distribution shift problems.

## 2.2 HIERARCHICAL DEPENDENCY CONTROLLED LEARNING FOR DYNAMIC DEPENDENCIES

To address the dynamic and multiscale temporal dependencies, we propose a Dependency Controlled Learning (DCL) module that adaptively controls the receptive field at the latent representation (Figure 3b). Temporal dependencies vary in temporal range, which requires the model to capture both short- and long-range patterns. Our multi-scale design enables the model to flexibly adjust the dependency range and effectively capture multi-scale patterns across varying temporal structures.

Specifically, given the denoised positive pair $(\mathcal{X}_p, \tilde{\mathcal{X}}_p)$ (discussed in next section), we omit the modeling of $\tilde{\mathcal{X}}_p$ as both inputs share the same encoder backbone. The input patches $\mathcal{X}_p \in \mathbb{R}^{N \times P}$ are first projected into a latent space via a linear transformation:

$$\mathcal{Z}_p = \mathcal{X}_p W_P + Bias, \quad \mathcal{Z}_p = \mathcal{Z}_p + W_{pos}, \tag{8}$$

where $W_P \in \mathbb{R}^{P \times D}$, $Bias \in \mathbb{R}^D$ and $D$ is the model dimension. To retain the temporal structure, we incorporate a learnable relative positional encoding $W_{pos} \in \mathbb{R}^{N \times D}$ into the latent representation $\mathcal{Z}_p$. To capture dependencies across different temporal scales, we define a set of window sizes $\{W_k\}_{k=1}^K$ for hierarchical modeling. For each window size, we reshape the $\mathcal{Z}_p$ into a windowed form and flatten the local window dimensions.

$$\mathcal{Z}_r = Flatten(Reshape(Padding(\mathcal{Z}_p))). \tag{9}$$

Specifically, $Z_p$ is first padded and reshaped into the defined window size $W_k$, converting from $\mathbb{R}^{N \times D}$ to $\mathbb{R}^{N/W_k, W_k, D}$, and then flatten the last two dimensions into $\mathbb{R}^{N/W_k, W_k \times D}$. This enables the encoder to operate over local windows of varying sizes. Then, the resulting representation within each window is passed to a temporal learner $\mathcal{E}_{enc}$:

$$\mathcal{Z}_e = ReshapeBack(\mathcal{E}_{enc}(\mathcal{Z}_r)), \tag{10}$$

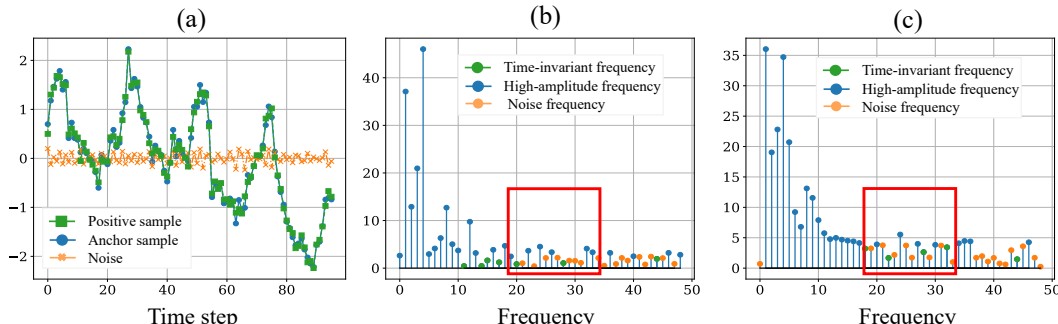

Figure 4: The ICM module filters time-variant noise while preserving time-invariant frequencies. (a) Anchor sample, filtered noise, and generated positive on ETTh1 (sequence length = 100), where the filtered noise (orange) resembles zero-mean white noise. (b) Amplitude spectrum with green dots for retained frequencies and orange dots for those removed by FMask. (c) Green-highlighted frequencies remain prominent, confirming their stability and importance for capturing patterns.

where $\mathcal{E}_{enc}$ is based on a Linear or MLP structure in our experiment. After that, we reshape the input $\mathcal{Z}_e$ to its original size $\mathbb{R}^{N \times D}$ and leverage residual connections to ensure training stability:

$$\mathcal{Z}_k = \mathcal{Z}_e + Dropout(\mathcal{Z}_p). \tag{11}$$

The window size $W_k$ increases across encoder blocks, enabling a global receptive view through careful design. This scale-aware structure empowers DCL to adaptively learn temporal dependencies from both local and global perspectives, making it suitable for handling varying and evolving temporal structures in time series.

## 2.3 Instance-level Contrastive Module (ICM) for Global Semantics

We introduce ICM to enhance DCL for global semantic modeling, as MTM paradigm may be suboptimal for high-level downstream tasks such as classification compared to CL paradigm (Figure 3a and Figure 4). To construct positive sample pairs for DCL's high-level semantic modeling, we incorporate a controllable time-variant low-amplitude filtering to generate noise-less views. Specifically, we first apply a Discrete Fourier Transform $\mathcal{F}$ to the batched input $\mathcal{X}_b = [\mathcal{X}_p^0, \mathcal{X}_p^1, ..., \mathcal{X}_p^M] \in \mathbb{R}^{B,M,N}$, where $B$ denotes batch size, $M$ denotes channel size, resulting in the corresponding spectrum: $S_b = \mathcal{F}(X_b)$. where the generated $\mathcal{S}_b$ has shape $[B, M, \lfloor \frac{L}{2} \rfloor]$. To identify stable global frequency components, $\mathcal{S}_{\text{mean}}$ across the batch and channel dimensions, we compute the average amplitude:

$$\mathcal{S}_{\text{mean}} = \text{AVG}(\text{AVG}(\text{AMP}(\mathcal{S}_b), \dim = 0), \dim = 0), \tag{12}$$

where AMP denotes the calculation of amplitude, and AVG represents the average operation. We then apply a $\text{top}K$ filter to extract global high-amplitude frequencies, forming the time- and channel-invariant set $\mathcal{S}_{\text{invar}}$. Next, a $\text{top}M$ filter selects instance-specific low-amplitude frequencies to generate the time-variant set $\mathcal{S}_{\text{var}}$. The selection process is defined as:

$$\mathcal{S}_{\text{invar}} = \text{Filter}(\text{top}K, \mathcal{S}_{\text{mean}}), \mathcal{S}_{\text{var}} = \text{Filter}(\text{top}M, \text{AMP}(\mathcal{S}_b)), \tag{13}$$

where $\text{top}K = \beta \times \lfloor L/2 \rfloor$, $\beta$ is a hyperparameter that controls the filtering intensity, and $\text{top}M = (1 - \beta) \times \lfloor L/2 \rfloor$. To prevent information loss, the final time-variant filtered $\mathcal{S}_{\text{var}}$ is defined as:

$$\mathcal{S}_{\text{var}} = \mathcal{S}_{\text{var}} - \mathcal{S}_{\text{var}} \cap \mathcal{S}_{\text{invar}}. \tag{14}$$

We construct a binary frequency mask $\text{FMask} \in \{0,1\}^{B \times M \times \lfloor L/2 \rfloor}$ to suppress time-variant components. Specifically, each element $\text{FMask}_{b,m,k}$ is set to 0 if $i \in \mathcal{S}_{\text{var}}$, and 1 otherwise, where $k \in [0, \lfloor L/2 \rfloor)$, $b \in [1, B]$, and $m \in [1, M]$. FMask is applied to obtain a filtered spectrum. The denoised signal $\tilde{X}_b$ is then recovered by applying the inverse Fourier transform $\mathcal{F}^{-1}$:

$$\tilde{X}_b = \mathcal{F}^{-1}(\text{FMask} \odot \mathcal{S}_b), \tag{15}$$

where $\odot$ denotes the Hadamard product. Once the batched positive pairs $(\mathcal{X}_b, \tilde{\mathcal{X}}_b)$ are obtained, where $\tilde{\mathcal{X}}_b$ preserves reliable temporal structures, we apply random masking $m$ to each pair $(\mathcal{X}_p, \tilde{\mathcal{X}}_p)$

Table 1: Forecasting results for predicting $F$ future time points based on the past 512 points in an in-domain setting. Results are averaged over $F \in \{96, 192, 336, 720\}$, with lower MSE and MAE indicating better performance. The best and second-best results are highlighted in **bold** and underlined, respectively.

| Model | ETTh1 | | ETTh2 | | ETTm1 | | ETTm2 | | Weather | | Electricity | | **Average** | |
|---|---|---|---|---|---|---|---|---|---|---|---|---|---|---|
| | MSE | MAE | MSE | MAE | MSE | MAE | MSE | MAE | MSE | MAE | MSE | MAE | MSE | MAE |
| Informer | 1.033 | 0.799 | 3.303 | 1.439 | 0.872 | 0.691 | 1.305 | 0.797 | 0.568 | 0.522 | 0.329 | 0.415 | 1.235 | 0.777 |
| Autoformer | 0.473 | 0.477 | 0.422 | 0.443 | 0.515 | 0.493 | 0.310 | 0.357 | 0.335 | 0.379 | 0.214 | 0.327 | 0.378 | 0.412 |
| Fedformer | 0.428 | 0.454 | 0.388 | 0.434 | 0.382 | 0.422 | 0.292 | 0.343 | 0.310 | 0.357 | 0.207 | 0.321 | 0.335 | 0.389 |
| iTransformer | 0.479 | 0.477 | 0.387 | 0.418 | 0.371 | 0.400 | 0.272 | 0.333 | 0.246 | 0.278 | 0.161 | 0.256 | 0.319 | 0.360 |
| DLinear | 0.433 | 0.446 | 0.477 | 0.469 | 0.360 | 0.384 | 0.283 | 0.346 | 0.247 | 0.311 | 0.162 | 0.260 | 0.327 | 0.369 |
| TimeMixer | 0.432 | 0.446 | 0.375 | 0.413 | 0.395 | 0.389 | 0.262 | 0.322 | 0.228 | 0.269 | 0.165 | 0.261 | 0.309 | 0.353 |
| CycleNet | 0.430 | 0.440 | 0.367 | 0.406 | 0.368 | 0.395 | 0.267 | 0.325 | 0.224 | 0.265 | 0.158 | 0.252 | 0.302 | 0.347 |
| PatchTST | 0.430 | 0.445 | 0.355 | 0.394 | 0.346 | 0.383 | 0.257 | 0.318 | 0.225 | 0.261 | **0.157** | 0.252 | 0.295 | 0.342 |
| SimMTM | 0.404 | 0.428 | 0.348 | 0.391 | 0.362 | 0.393 | 0.269 | 0.327 | 0.227 | 0.268 | 0.162 | 0.256 | 0.295 | 0.344 |
| **DeCoP$_{\text{Linear}}$** | **0.401** | **0.421** | **0.333** | **0.382** | 0.361 | 0.379 | 0.255 | 0.313 | 0.242 | 0.279 | 0.165 | 0.258 | 0.293 | 0.339 |
| **DeCoP$_{\text{MLP}}$** | 0.408 | 0.424 | 0.341 | 0.388 | **0.342** | **0.376** | **0.249** | **0.311** | **0.223** | **0.259** | **0.157** | **0.251** | **0.287** | **0.335** |

from the batch for subsequent masked modeling. For each positive pair, the temporal learner $\mathcal{E}_{enc}$ extracts their latent representations as:

$$\mathcal{Z}_e = \mathcal{E}_{enc}(\mathcal{X}_p), \quad \tilde{\mathcal{Z}}_e = \mathcal{E}_{enc}(\tilde{\mathcal{X}}_p). \tag{16}$$

We further define a similarity-based contrastive loss function $\mathcal{L}_{cl}$ for the final DCL block as follows:

$$\mathcal{L}_{cl} = 1 - \frac{1}{BM} \sum_{i=1}^{B} \sum_{m=1}^{M} \text{AVG}(\mathcal{Z}_e) \cdot \text{AVG}(\tilde{\mathcal{Z}}_e), \tag{17}$$

where $\cdot$ denotes the dot product. By minimizing $\mathcal{L}_{cl}$, the model encourages $\mathcal{Z}_e$ to approximate its denoised counterpart $\tilde{\mathcal{Z}}_e$, enhancing global instance-level representation learning and improving generalization to downstream tasks.

## 2.4 LOSS DESIGN

Based on the MTM framework, we reconstruct masked patches from the unmasked ones. A pre-training head predicts the masked patches as $\hat{\mathcal{X}}_p = \mathcal{Z}_k W + \text{Bias}$, where $W \in \mathbb{R}^{D \times P}$, $\text{Bias} \in \mathbb{R}^P$, and $\mathcal{Z}_k \in \mathbb{R}^{N \times D}$. The reconstruction loss is computed using MSE:

$$\mathcal{L}_{recon} = \sum_{i=1}^{B} \sum_{m=1}^{M} \sum_{n=1}^{N} \left\| m \odot (\mathcal{X}_p^{(i,m,n)} - \hat{\mathcal{X}}_p^{(i,m,n)}) \right\|_2^2, \tag{18}$$

where $m$ is a binary mask indicating whether a patch is masked and $\odot$ denotes the Hadamard product. The final loss combines the reconstruction loss with the contrastive loss previously introduced:

$$\mathcal{L} = \mathcal{L}_{recon} + \gamma \times \mathcal{L}_{cl}, \tag{19}$$

where the hyperparameter $\gamma$ is the weight of $\mathcal{L}_{cl}$. This combined loss enhance temporal consistency under temporal noise and non-uniform dependencies, encouraging the model to learn representations that preserve global patterns.

## 3 EXPERIMENTS

**Experiment Setting**. We conduct experiments on forecasting and classification tasks, following the protocols in Nie et al. (2022) and Dong et al. (2024). Fine-tuning performance is evaluated under in-domain and cross-domain settings. MSE and MAE are used as metrics for forecasting, while Accuracy, Precision, Recall, and F1-score assess classification performance. For forecasting, six real-world datasets are employed, including four ETT datasets Zhou et al. (2021) (ETTh1, ETTh2, ETTm1, ETTm2), Weather Wetterstation (2021), and Electricity UCI (2021). For classification, we adopt four real-world datasets: SleepEEG Kemp et al. (2000), Epilepsy Andrzejak et al. (2001), FD-B Lessmeier et al. (2016), and EMG PhysioBank (2000).

**Model Parameters**. By default, all experiments are configured with the following parameters: $k = 2$, $\text{top}K = 0.3$, $\alpha_{initial} = 0.01$, $\gamma = 0.1$ and $lr = 1e - 4$. For forecasting tasks, both in-

Table 2: Transfer learning setting of forecasting the future $F$ time points. All results are averaged from 4 different choices of $F \in \{96, 192, 336, 720\}$. The best and second-best results are highlighted in **bold** and underlined, respectively.

| Scenarios | | DeCoP$_{Linear}$ | | DeCoP$_{MLP}$ | | PatchTST | | SimMTM | | TimeMAE | | CoST | | TST | | TF-C | |
|---|---|---|---|---|---|---|---|---|---|---|---|---|---|---|---|---|---|
| Source | Target | MSE | MAE | MSE | MAE | MSE | MAE | MSE | MAE | MSE | MAE | MSE | MAE | MSE | MAE | MSE | MAE |
| ETTh2 | ETTh1 | **0.403** | **0.422** | 0.409 | 0.426 | 0.423 | 0.443 | 0.415 | 0.43 | 0.466 | 0.456 | 0.428 | 0.433 | 0.469 | 0.459 | 0.635 | 0.634 |
| ETTm2 | ETTm1 | 0.359 | 0.379 | **0.342** | **0.376** | 0.348 | 0.382 | 0.351 | 0.383 | 0.390 | 0.410 | 0.385 | 0.412 | 0.382 | 0.402 | 0.758 | 0.669 |
| ETTm2 | ETTh1 | **0.404** | **0.423** | 0.412 | 0.426 | 0.433 | 0.447 | 0.428 | 0.441 | 0.464 | 0.456 | 0.598 | 0.548 | 0.453 | 0.45 | 1.091 | 0.814 |
| ETTh2 | ETTm1 | 0.360 | 0.379 | **0.343** | **0.377** | 0.363 | 0.387 | 0.365 | 0.384 | 0.383 | 0.402 | 0.363 | 0.387 | 0.391 | 0.409 | 0.750 | 0.654 |
| ETTm1 | ETTh1 | **0.405** | **0.423** | 0.416 | 0.427 | 0.447 | 0.451 | 0.422 | 0.430 | 0.495 | 0.469 | 0.62 | 0.541 | 0.475 | 0.463 | 0.700 | 0.702 |
| ETTh1 | ETTm1 | 0.361 | 0.379 | **0.346** | **0.379** | 0.348 | 0.381 | 0.346 | 0.384 | 0.360 | 0.390 | 0.37 | 0.393 | 0.373 | 0.393 | 0.746 | 0.652 |
| Weather | ETTh1 | **0.405** | **0.422** | 0.411 | 0.426 | 0.437 | 0.448 | 0.456 | 0.467 | 0.518 | 0.487 | 0.465 | 0.456 | 0.462 | 0.464 | - | - |
| Weather | ETTm1 | 0.359 | 0.379 | **0.345** | **0.376** | 0.348 | 0.383 | 0.358 | 0.388 | 0.411 | 0.423 | 0.382 | 0.403 | 0.368 | 0.392 | - | - |
| **Average** | | 0.382 | **0.401** | **0.378** | 0.402 | 0.393 | 0.415 | 0.393 | 0.413 | 0.429 | 0.434 | 0.458 | 0.451 | 0.422 | 0.428 | 0.780 | 0.693 |

Table 3: In- and cross-domain classification. For in-domain, DeCoP is pretrained and finetuned on Epilepsy. For the cross-domain setting, we pretrain DeCoP on SleepEEG and fine-tune it to multiple target datasets: Epilepsy, FD-B, EMG. P and R denotes precision and recall, respectively. The best and second-best results are highlighted in **bold** and underlined, respectively.

| Scenarios | In-domain | | | | Cross-domain | | | | | | | | | | | |
|---|---|---|---|---|---|---|---|---|---|---|---|---|---|---|---|---|
| | Epilepsy → Epilepsy | | | | SleepEEG → Epilepsy | | | | SleepEEG → FD-B | | | | SleepEEG → EMG | | | |
| Metrics | ACC | P | R | F1 | ACC | P | R | F1 | ACC | P | R | F1 | ACC | P | R | F1 |
| TS2Vec | 92.17 | 93.84 | 81.19 | 85.71 | 93.95 | 90.59 | 90.39 | 90.45 | 47.9 | 43.39 | 48.42 | 43.89 | 78.54 | 80.4 | 67.85 | 67.66 |
| LaST | 92.11 | 93.12 | 81.47 | 85.74 | 86.46 | 90.77 | 66.35 | 70.67 | 46.67 | 43.9 | 47.71 | 45.17 | 66.34 | 79.34 | 63.33 | 72.55 |
| TF-C | 93.96 | 94.87 | 85.82 | 89.46 | 94.95 | **94.56** | 80.08 | 91.49 | 69.38 | 75.59 | 72.02 | 74.87 | 81.71 | 72.65 | 81.59 | 76.83 |
| TST | 80.21 | 40.11 | 50.00 | 44.51 | 80.21 | 40.11 | 50.00 | 44.51 | 46.4 | 41.58 | 45.5 | 41.34 | 78.34 | 77.11 | 80.3 | 68.89 |
| CoST | 88.07 | 91.58 | 66.05 | 69.11 | 88.40 | 88.20 | 72.34 | 76.88 | 47.06 | 38.79 | 38.42 | 34.79 | 53.65 | 49.07 | 42.1 | 35.27 |
| Ti-MAE | 90.90 | 93.90 | 77.24 | 78.21 | 89.71 | 72.36 | 67.47 | 68.55 | 60.88 | 66.98 | 68.94 | 66.56 | 69.99 | 70.25 | 63.44 | 70.89 |
| PatchTST | 89.56 | 90.39 | 89.56 | 80.11 | 93.27 | 92.51 | 85.57 | 88.48 | 80.15 | 82.25 | 85.47 | 83.05 | 90.24 | 82.96 | 82.95 | 82.91 |
| SimMTM | 94.75 | 95.6 | 89.93 | 91.41 | 95.49 | 93.36 | 92.28 | 92.81 | 69.40 | 74.18 | 76.41 | 75.11 | 97.56 | 98.33 | 98.04 | 98.14 |
| **DeCoP$_{MLP}$** | **95.53** | 93.51 | **92.25** | **92.86** | **95.82** | 94.23 | **92.41** | **93.28** | **93.04** | **94.92** | **94.90** | **94.90** | **100** | **100** | **100** | **100** |

and cross-domain experiments share the same configuration, with a patch size and stride of 12. For both in- and cross-domain classification task, the patch size and stride are set to 8. More details of parameters are provided in appendix.

## 3.1 TIME SERIES FORECASTING

**In-domain Evaluation**. We compare our model with six competitive state-of-the-art baseline methods in time series forecasting, including self-supervised approaches (SimMTM Dong et al. (2024), PatchTST Nie et al. (2022)) and supervised approaches (CycleNet Lin et al. (2024), TimeMixer Wang et al. (2024), DLinear Zeng et al. (2023), iTransformer Liu et al. (2023), Fedformer Zhou et al. (2022), Autoformer Wu et al. (2021), Informer Zhou et al. (2021)). The look-back period to 512, with a patch size of 12 and a stride of 12 across all forecasting experiments. The patches remain non-overlapping during both the pre-training and fine-tuning stages. In Table 1, DeCoP$_{Linear}$ outperforms the second-best by 1.5% on ETTh2. For more complex datasets like ETTm2, DeCoP$_{MLP}$ achieves the best results, surpassing the PatchTST by 0.8% in MSE.

**Cross-domain Analysis**. In the cross-domain setting, we compare our framework with six advanced time series pre-training frameworks SimMTM, PatchTST, TF-C Zhang et al. (2022a), TST Zerveas et al. (2021), CoST Woo et al. (2022) and TimeMAE Li et al. (2023)). In Table 2, we evaluate multiple scenarios to test effectiveness under cross-domain conditions. Both in-domain and cross-domain transfer settings, our model consistently achieves lower MSE and MAE than others, especially in ETTm1 → ETTh1, we outperform PatchTST 4.2% in MSE, highlighting its effectiveness under distribution shifts. Complete forecasting results are provided in the appendix.

## 3.2 TIME SERIES CLASSIFICATION

**In-domain Evaluation**. For in-domain learning, We preform Epilepsy → Epilepsy following Dong et al. (2024). We adopt MLP as our temporal learner in classification task and compare it with eight competitive state-of-the-art baseline methods, including the contrastive learning based methods: TF-C, LaST, TST, TS2Vec Yue et al. (2022), and the masked time series modeling methods: SimMTM, PatchTST, Ti-MAE, CoST Wang et al. (2022). In Table 3, Our model outperform second-best SimMTM by 1.45% in F1, and outperform PatchTST by 12.75%.

Table 4: The left table compares FLOPs, parameters, and average MAE on the ETTh1 and Weather datasets across different pre-training frameworks. The right table illustrates the effect of controllable window sizes $W_k$, which enable efficient pretraining by allowing flexible dependency modeling.

| Dataset | Models | Pretrain | | Finetune | | MSE |
|---------|--------|----------|--------|----------|--------|-----|
| | | FLOPs | Params | FLOPs | Params | |
| ETTh1 | PatchTST | 175M | 0.598M | 130M | **2M** | 0.430 |
| | SimMTM | 4269M | 143M | 100M | 6M | 0.404 |
| | **DeCoP** | **72M** | **0.479M** | **49M** | 2M | **0.401** |
| Weather | PatchTST | 526M | 0.598M | 389M | **2M** | 0.225 |
| | SimMTM | 48865M | 556M | 259M | 11M | 0.227 |
| | **DeCoP** | **245M** | **0.463M** | **159M** | 2M | **0.223** |

| $W_k$ | ETTh1 | | ETTh2 | | ETTm1 | | Params |
|-------|-------|-----|-------|-----|-------|-----|--------|
| | MSE | MAE | MSE | MAE | MSE | MAE | |
| 1,1 | 0.406 | 0.424 | 0.335 | 0.383 | 0.348 | 0.382 | 0.165M |
| 1,3 | 0.403 | 0.422 | 0.335 | 0.384 | 0.347 | 0.381 | 0.446M |
| 2,5 | **0.401** | **0.421** | **0.333** | **0.382** | 0.346 | 0.377 | 0.999M |
| 4,8 | 0.403 | 0.423 | 0.337 | 0.385 | **0.342** | **0.376** | 2.3M |
| 42,42 | 0.405 | 0.423 | 0.335 | 0.383 | 0.345 | 0.377 | 88.8M |

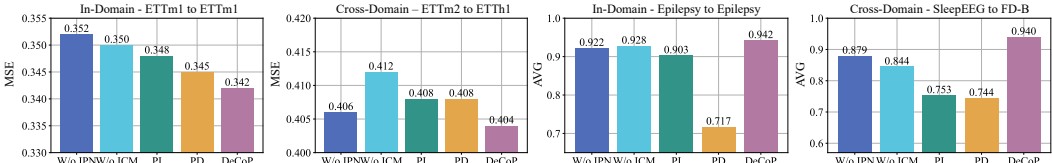

Figure 5: Ablation study of DeCoP, showing the impact of IPN, ICM, DCL, Patch Dependence, and Patch Independence on time series forecasting (left) and classification (right) tasks in both in- and cross-domain settings. For classification, AVG denotes the average of accuracy and F1 score.

**Cross-domain Analysis**. For cross-domain setting, we conduct experiments across in-domain and cross-domain transfer learning SleepEEG $\rightarrow$ {Epilepsy, FD-B, EMG} in Table 3, where the source data differs from the target data in both channels and classes. Notably, on SleepEEG $\rightarrow$ FD-B, Our framework surpass the second best by 12.89% and 11.85% in accuracy and F1, respectively. These results highlight DeCoP's superior robustness under both domain and label shifts.

### 3.3 MODEL ANALYSIS

#### 3.3.1 ABLATION STUDIES

We conduct ablation studies on both forecasting and classification tasks under in-domain and cross-domain settings to evaluate the contributions of IPN, ICM, and DCL. For DCL, we compare two alternative configurations: patch-independent (PI) and patch-dependent (PD). In Figure 5, replacing DCL with PI or PD leads to average performance drops of 3.94% and 22.51% on in-domain classification, respectively. The performance gap becomes larger in the cross-domain setting, with declines of 18.69% (PI) and 19.57% (PD), highlighting the importance of dynamical dependency modeling. In forecasting, removing IPN and ICM results in MSE increases of 0.5% and 0.4% under in-domain and cross-domain settings, respectively.

Notably, ICM contributes to more stable gains across both forecasting and classification: its removal increases forecasting error by up to 0.8% in the ETTm2 $\rightarrow$ ETTh1 task and reduces the average of F1 and accuracy by 9.53% in the SleepEEG $\rightarrow$ FD-B scenario. These results confirm the effectiveness of ICM in enhancing generalization to downstream tasks, particularly for high-level classification tasks. More ablation results are provided in appendix.

#### 3.3.2 BETTER RESULTS WITH COMPACT MODEL

We compute the FLOPs and parameters of DeCoP compared to two SOTA pre-training frameworks in two datasets in the left table of Table 4. In both pre-training and fine-tuning stages, DeCoP achieves the lowest MSE 0.401, outperforming PatchTST by 30% on the ETTh1 dataset while using only 37% of the FLOPs. Full results on efficiency are provided in appendix.

#### 3.3.3 LEVERAGING PERIODICITY PRIORS FOR SUPERIOR EFFICIENCY

DeCoP outperforms prior methods such as PatchTST and SimMTM, achieving superior performance with fewer parameters through a controllable modeling mechanism. The DCL module captures dependencies between patches using variable window sizes, enabling alignment with periodic patterns in time series data. Empirically, we adopt a $(2, *)$ configuration to capture daily periodicity in hourly datasets with a patch size of 12, and a $(4, *)$ configuration to capture hourly periodicity in 10-minute

datasets. In the right table of Table 4, The $(2, 5)$ setting achieves strong performance with only 999k parameters, highlighting the efficiency of DCL.

### 3.3.4 ADVANCING CONTROLLABILITY WITH DeCoP

A key challenge in time series pre-training is modeling temporal dependencies that evolve due to distribution shifts (Figure 6a) and multi-scale patterns, often resulting in noisy features and poor generalization. While existing approaches like PatchTST employ instance-level normalization (IN) to mitigate distribution shifts, we observe that IN tends to oversmooth patch-level variations, weakening semantic expressiveness (Figure 6b). In contrast, DeCoP explicitly addresses this challenge through controllable normalization. IPN adaptively normalizes both fine-grained patch-level statistics and coarse-grained instance-level distributions. This dual-level normalization allows the model

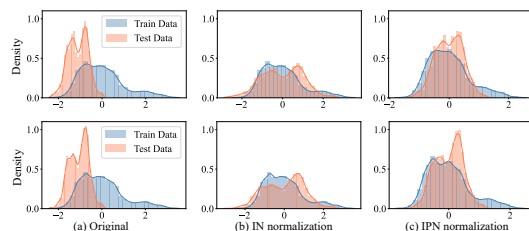

Figure 6: Comparison of data distributions before normalization (a), after IN (b), and after IPN (c) on ETTh1 and ETTm1. IPN preserves original semantic patterns such as peaks while better aligning train and test distributions.

to better preserve local temporal semantics while maintaining global statistical alignment. Compared to IN, IPN more effectively retains informative intra-patch variations (Figure 6c).

Additionally, our DCL method controllably encodes temporal structures through dynamic window grouping and encoding dependencies hierarchically. Unlike single scale attention in PatchTST, DCL explicitly constrains the temporal scope of dependency modeling, allowing the model to capture meaningful local patterns and gradually expand to global semantics. This controllable design reduces overfitting risks under distribution shifts by avoiding noisy or ir-

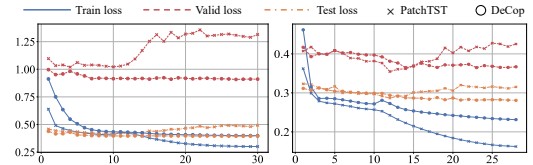

Figure 7: DeCoP achieves faster convergence and smaller train–val loss gaps on ETT datasets.

relevant dependencies. In Figure 7, DeCoP converges faster and maintains a smaller gap between training and validation loss, demonstrating better generalization and reduced overfitting.

### 3.3.5 ROBUST GENERALIZATION UNDER LIMITED DATA

We further assess DeCoP's generalization capability under limited data scenarios in Table 5. In the ETTh2→ETTh1 transfer setting with only 50% labeled data, DeCoP outperforms PatchTST and SimMTM by 2.2% and 4.9% in MSE, respectively. DeCoP consistently achieves the lowest MSE and MAE across all finetuning ratios (25%, 50%, and 75%), highlighting its robustness in data-scarce scenarios.

Table 5: Transfer performance from ETTh2 to ETTh1 under different finetuning ratios.

| ETTh2→ETTh1 | 25% | | 50% | | 75% | |
|---|---|---|---|---|---|---|
| **Models** | **MSE** | **MAE** | **MSE** | **MAE** | **MSE** | **MAE** |
| SimMTM | 0.468 | 0.469 | 0.451 | 0.461 | 0.428 | 0.445 |
| PatchTST | 0.453 | 0.462 | 0.424 | 0.440 | 0.425 | 0.441 |
| DeCoP | **0.445** | **0.457** | **0.402** | **0.423** | **0.405** | **0.423** |

## 4 CONCLUSION

This paper introduces **DeCoP**, a Dependency Controlled Pretraining framework that improve time series representation learning by explicitly modeling dynamic and multi-scale temporal dependencies. At the input level, IPN establishes a stable foundation by mitigating distribution shifts through instance-wise patch normalization while preserving fine-grained, patch-level information. At the latent representation level, DCL explicitly captures multi-scale dependencies through controllable receptive filed and ICM enhances global representation learning by incorporating time-invariant positive pairs. DeCoP outperforms existing models with fewer parameters, highlighting the importance of dependency-controlled pre-training for dynamic time series. We hope that DeCoP can inspire future research in building more general, efficient, and controllable pre-training paradigms.

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
