## A  THE USE OF LARGE LANGUAGE MODELS (LLMs)

We used LLMs for language polishing and minor rewrites of paragraphs after we had written the technical content. We did not use LLMs to generate research ideas, design experiments, or write technical sections. All analyses, methods, and results were conceived and verified by the authors.

## B  OVERVIEW

We first present a comprehensive review in the Section C section and a detailed overview of the Section D, including descriptions of the datasets and baseline models. We then evaluate DeCoP's Section E in terms of computational efficiency, generalization under limited data conditions, and the effects of different filtering strategies in the Instance-level Contrastive Module (ICM). Next, we perform a Section F, investigating the impact of varying patch lengths and the learnable hyperparameter $\alpha_{\text{initial}}$ for the Instance-wise Patch Normalization (IPN) module, different look-back lengths for the Dependency Controlled Learning (DCL) module, and varying filter intensities $\beta$ and contrastive loss weights $\gamma$ for ICM. Finally, we provide the Section G and the Section H for both time series forecasting and classification tasks.

## C  RELATED WORK

### C.1  SELF-SUPERVISED LEARNING

Self-supervised learning (SSL) has become a dominant paradigm across domains, with notable examples including Masked Language Modeling (MLM) Devlin et al. (2019) and Generative Pre-trained Models (GPM) Brown et al. (2020). In MLM, random tokens are masked in text and predicted based on surrounding unmasked tokens, while GPM predicts the next token in an autoregressive manner. These methods leverage large unlabeled datasets, allowing models to learn meaningful representations without manual labeling, which supports scalable learning across vast datasets, preserves data diversity, and minimizes labeling costs. Contrastive learning (CL) Chen et al. (2020a); Gao et al. (2021) has also gained attraction, focusing on maximizing similarity between positive pairs while minimizing it between negative pairs. Foundational works such as Sim-CLR Chen et al. (2020b) and MoCo He et al. (2020) in computer vision, along with CLIP Radford et al. (2021) in multimodal alignment, underscore its versatility. However, our framework combines the self-supervised nature of MLM with the contrastive principles of CL, enhancing robustness and consistency in feature learning to address distribution shifts.

### C.2  MASKED TIME SERIES MODELING

Inspired by the success of Masked Language Modeling (MLM), masked time series modeling (MTM) Rasul et al. (2023); Garza et al. (2023); Das et al. (2023) has gained popularity in time series analysis. PatchTST Nie et al. (2022) first introduced the patching technique and masked modeling pretext task for low-level time series forecasting. SimMTM Dong et al. (2024) constructed positive samples from a manifold perspective and reconstructed time series from multiple masked sequences, while needing large computing resources when training. Meanwhile, CL has been widely adopted for high-level time series classification tasks. TF-C Zhang et al. (2022b) proposed a time and frequency domain contrastive learning framework to enhance consistency in both the time and frequency domains. In contrast, our framework better at handling complex distributions and dynamical dependency by incorporating dependence controlled learning.

## D  EXPERIMENTAL DETAILS

### D.1  DATASET

We evaluate our framework using 10 datasets across forecasting and classification tasks in both in-domain and cross-domain settings. Detailed descriptions of the datasets are provided in Table 6. The ETT datasets Zhou et al. (2021) (ETTh1, ETTh2, ETTm1, and ETTm2) were collected from two distinct electric transformers over a two-year period, from July 2016 to July 2018. These datasets are

available in two temporal resolutions: 15 minutes and 1 hour, denoted as "m" and "h," respectively. The Weather dataset Wetterstation (2021) comprises 21 meteorological indicators recorded every 10 minutes in Germany during 2020. The Electricity dataset UCI (2021) contains hourly electricity consumption records of 321 customers from 2012 to 2014.

For classification tasks, the SleepEEG dataset Kemp et al. (2000) includes 153 whole-night electroencephalography (EEG) recordings, categorized into five stages: Wake (W), Non-rapid eye movement (N1, N2, N3), and Rapid Eye Movement (REM). The EPILEPSY dataset Andrzejak et al. (2001) features single-channel EEG measurements from 500 subjects, with binary labels indicating whether the subject experienced a seizure. The FD-B dataset Lessmeier et al. (2016) was generated using an electromechanical drive system to monitor rolling bearing conditions and classify faults into three categories:

Table 6: Datasets for Forecasting and Classification Tasks

| Tasks | Datasets | Channels | Length | Classes | Frequency |
|---|---|---|---|---|---|
| Forecasting | ETTh1 | 7 | 17420 | - | 1 Hour |
|  | ETTh2 | 7 | 17420 | - | 1 Hour |
|  | ETTm1 | 7 | 69680 | - | 15 Mins |
|  | ETTm2 | 7 | 69680 | - | 15 Mins |
|  | Weather | 21 | 52696 | - | 10 Mins |
|  | Electricity | 321 | 26304 | - | 1 Hour |
| Classification | SleepEEG | 1 | 200 | 5 | 100 Hz |
|  | Epilepsy | 1 | 178 | 2 | 174 Hz |
|  | FD-B | 1 | 5120 | 3 | 64K Hz |
|  | EMG | 1 | 1500 | 3 | 4K Hz |

undamaged, inner-damaged, and outer-damaged. The Electromyogram (EMG) dataset PhysioBank (2000) records electrical activity in muscle responses to neural stimulation. It consists of single-channel EMG recordings from the tibialis anterior muscle of three healthy volunteers suffering from neuropathy and myopathy, where each patient represents a classification category.

## D.2 DETAILS OF BASELINE SETTINGS

For the time series forecasting task, we categorize the baseline models into two paradigms: self-supervised and supervised. PatchTST Nie et al. (2022) and SimMTM Dong et al. (2024) are representative self-supervised models. In contrast, DLinear Zeng et al. (2023), FEDformer Zhou et al. (2022), Autoformer Wu et al. (2021), and Informer Zhou et al. (2021) are robust supervised models for forecasting tasks. Additionally, CycleNet and TimeMixer represent the latest state-of-the-art methods, also grounded in the supervised paradigm. For the classification task, we divide the baseline models into two paradigms: masked time series models (MTM) and contrastive learning (CL) models. SimMTM Dong et al. (2024), Ti-MAE Li et al. (2023), and TST Zerveas et al. (2021) follow the MTM paradigm, while LaST Wang et al. (2022), TF-C Zhang et al. (2022a), CoST Woo et al. (2022), and TS2Vec Yue et al. (2022) are based on the CL paradigm.

For time series forecasting task, the default look-back window for various MTM models is set to 512, following Nie et al. (2022). The results for DLinear , FEDformer, Autoformer, and Informer are from PatchTST. Meanwhile, for time series classification task,the results for Ti-MAE, TST, LaST , TF-C, CoST, and TS2Vec are obtained from SimMTM. For the latest state-of-the-art methods Wang et al. (2024); Lin et al. (2024), the look-back window length is consistently fixed at 512, adhering to Nie et al. (2022). The classification results of PatchTST are reproduced using the official codebase, with hyperparameters further tuned based on the default settings to achieve optimal performance.

## D.3 IMPLEMENTATION DETAILS

All experiments were repeated five times, implemented using PyTorch, and conducted on an NVIDIA RTX 4090 GPU with 24GB of memory. The baselines were implemented based on their official repositories, adhering to the configurations specified in their original papers. For forecasting tasks, all datasets were chronologically split into training, validation, and test sets, with splitting ratios of 6:2:2 for the ETT datasets and 7:1:2 for the other datasets, as outlined in Wu et al. (2021). For classification tasks, the dataset splits followed the setup described in Zhang et al. (2022a). During pre-training, each model was typically trained for 100 epochs. This was followed by linear probing of the head for 10 epochs and fine-tuning the entire model for 20 epochs, in line with Nie et al. (2022).

Table 7: Forecasting Configuration and Classification Configuration

| Task | Dataset | $d_{model}$ | $W_k$ | d1 | d2 |
|------|---------|-------------|-------|----|----|
| | ETTh1/ETTh2 | 128 | (2,5) | 256 | 512 |
| Forecasting | ETTm1/ETTm2 | 128 | (4,8) | 256 | 512 |
| | Weather | 128 | (2,5) | 256 | 512 |
| | Electricity | 256 | (3,6) | 512 | 512 |

(a)

| Source Data | Target Data | $d$ | $W_K$ | $d_1$ | $d_2$ | Agg |
|-------------|-------------|-----|-------|-------|-------|-----|
| Epilepsy | Epilepsy | 128 | (3,6) | 256 | 512 | Avg |
| SleepEEG | Epilepsy | 128 | (2,5) | 256 | 512 | Avg |
| SleepEEG | FD-B | 128 | (2,5) | 128 | 256 | Avg |
| SleepEEG | EMG | 128 | (2,5) | 256 | 512 | Max |

(b)

## D.4 MODEL PARAMETERS

By default, all experiments are configured with the following parameters: $e_{layers} = 2$, top$\mathcal{K} = 0.3$, $\alpha_{intial} = 0.01$, and $\gamma = 0.1$. During pre-training, a dropout ratio of 0.2 is applied. For forecasting tasks, both in-domain and cross-domain experiments share the same configuration, with a patch size and stride of 12. For classification tasks, the patch size is set to 8 for all datasets. A learning rate of 1e-4 is applied across all tasks during the pre-training and fine-tuning stage. Additional key parameters for forecasting and classification are detailed in Table 7.

## D.5 RESULTS WITH DIFFERENT RANDOM SEEDS

To examine the robustness of our results, we train the supervised PatchTST model with 5 different random seeds: 1,2,3,4,5 and calculate the MSE and MAE scores with each selected seed. The mean and standard derivation of the results are reported in Table 8. The findings demonstrate that the variances in MSE and MAE across different random seeds are notably small, indicating the stability and robustness of the model. This consistency suggests that De-CoP's performance is not significantly affected by random initialization, reinforcing the reliability of its predictions across different experimental setups. Specifically, the robustness is evident across diverse datasets, including ETTh1, ETTh2, ETTm1, ETTm2, Weather, and Electricity, where the standard deviations are consistently low, regardless of the prediction length.

Table 8: Comparison of DeCoP under different random seed across different forecasting datasets.

| Datasets | Pred len | DeCoP MSE | DeCoP MAE |
|----------|----------|-----------|-----------|
| ETTh1 | 96 | 0.3604±0.0010 | 0.3900±0.0008 |
| | 192 | 0.3935±0.0010 | 0.4104±0.0007 |
| | 336 | 0.4173±0.0033 | 0.4275±0.0016 |
| | 720 | 0.4328±0.0015 | 0.4552±0.0009 |
| ETTh2 | 96 | 0.2672±0.0024 | 0.3324±0.0011 |
| | 192 | 0.3281±0.0019 | 0.3735±0.0034 |
| | 336 | 0.3530±0.0034 | 0.3974±0.0032 |
| | 720 | 0.3818±0.0024 | 0.4254±0.0022 |
| ETTm1 | 96 | 0.2809±0.0033 | 0.3395±0.0014 |
| | 192 | 0.3254±0.0017 | 0.3661±0.0006 |
| | 336 | 0.3532±0.0038 | 0.3867±0.0010 |
| | 720 | 0.4098±0.0018 | 0.4134±0.0015 |
| ETTm2 | 96 | 0.1630±0.0006 | 0.2546±0.0006 |
| | 192 | 0.2172±0.0004 | 0.2900±0.0002 |
| | 336 | 0.2661±0.0010 | 0.3239±0.0011 |
| | 720 | 0.3496±0.0008 | 0.3770±0.0010 |
| Weather | 96 | 0.1456±0.0005 | 0.1931±0.0007 |
| | 192 | 0.1897±0.0003 | 0.2363±0.0004 |
| | 336 | 0.2421±0.0001 | 0.2770±0.0008 |
| | 720 | 0.3152±0.0008 | 0.3300±0.0008 |
| Electricity | 96 | 0.1274±0.0001 | 0.2223±0.0001 |
| | 192 | 0.1457±0.0001 | 0.2359±0.0001 |
| | 336 | 0.1606±0.0003 | 0.2555±0.0004 |
| | 720 | 0.1949±0.0008 | 0.2889±0.0008 |

## D.6 PERFORMANCE VISUALIZATIONS

We provide qualitative comparisons on the ETTh1 dataset to illustrate the predictive behavior of each model. As shown in Figure 8, DeCoP achieves superior performance, reducing MAE by 2.4% and 0.7% compared to PatchTST and SimMTM, respectively. Visually, DeCoP more accurately follows both the overall trend and the fine-grained fluctuations of the target signal, demonstrating its advantage in modeling temporal dynamics.

## D.7 POSITIVE SAMPLE PAIRS VISUALIZATION

We present qualitative examples of positive sample pairs generated by the proposed Instance-level Contrastive Module (ICM) across six representative datasets, as shown in Figure 9. These visualizations illustrate how ICM preserves the global temporal structure of anchor samples while introducing controlled perturbations for contrastive learning. The generated positive samples (green) maintain the overall trends and periodic patterns of the anchor sequences (blue) by selectively filtering out

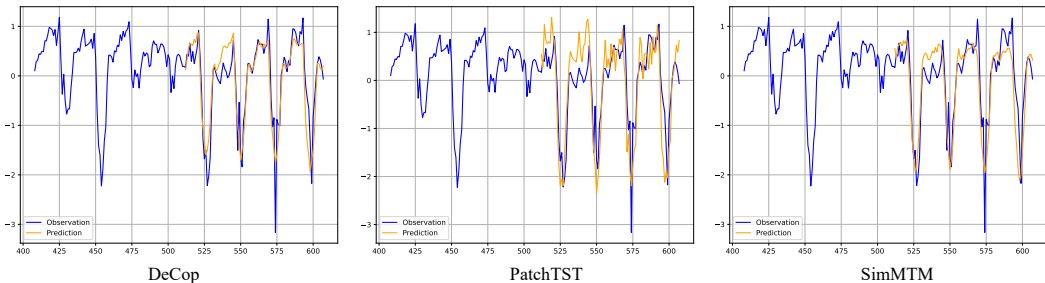

Figure 8: Performance visualization on the ETTh1 dataset. DeCoP captures both the overall trend and local fluctuations more accurately than PatchTST and SimMTM.

time-variant low-amplitude frequency components in the frequency domain. Compared to anchor sequences, the positive samples exhibit reduced noise and smoother trajectories, achieved by tuning the filtering intensity. Importantly, the filtered noise (orange) primarily consists of random fluctuations lacking meaningful temporal patterns, which are effectively suppressed in the positive samples.

These results demonstrate that ICM generates high-quality positive pairs that retain semantically important characteristics while attenuating irrelevant variations. When used with a contrastive loss, these samples enable DeCoP to learn more discriminative and generalized high-level representations from diverse time series inputs, improving generalization across various downstream tasks.

# E PERFORMANCE ANALYSIS

## E.1 EFFICIENCY

We evaluate DeCoP's efficiency for practical deployment. The ICM is removed during fine-tuning, incurring no impact on inference performance (see Table 9). During pretraining, ICM contributes minimally to resource consumption—accounting for only 10% of total training time and 6% of GPU memory (in MiB) per iteration across datasets. For example, on the ETTh1 dataset, ICM adds only 1.25 ms to pretraining

Table 9: Computation and memory costs across datasets.

| Metric | Pretrain Time | | Pretrain Mem | | Finetune Time | |
|---|---|---|---|---|---|---|
| Dataset | Overall | ICM | Overall | ICM | Overall | Inference |
| ETTh1 | 12.38 | 1.250 | 1072 | 6.36 | 4.23 | 1.82 |
| Weather | 17.05 | 1.678 | 1740 | 21.46 | 4.90 | 2.26 |
| Epilepsy | 16.32 | 2.30 | 536 | 0.31 | 4.72 | 1.03 |

time and 6.36 MiB of memory overhead. Despite its low cost, ICM remains effective, improving F1 by 9.95% in the SleepEEG→FD-B transfer scenario. For deployment, the IPN serves as a lightweight normalization layer, while the DCL module leverages simple temporal learners with low parameter overhead. As shown in Table 5 in main text, DeCoP consistently achieves lower inference FLOPs and latency than all baselines.

## E.2 GENERALIZATION ON ANOMALY DETECTION BENCHMARK

Similarly, to evaluate its generalization capabilities on other tasks, we benchmarked DeCoP on three anomaly detection datasets: SMAP Hundman et al. (2018), PSM Abdulaal et al. (2021), and MSL Hundman et al. (2018). As shown in Table 10a, DeCoP achieves state-of-the-art performance on two of the three datasets. Specifically, it obtains the highest F1 score on SMAP (87.80), outperforming the strong PatchTST baseline by 1.74%, and also leads on PSM with an F1 score of 94.86. On the MSL dataset. Overall, these results demonstrate DeCoP's robust generalization to anomaly detection tasks.

## E.3 ALTERNATIVE FILTER METHODS

To test the effectiveness of our time-invariant filter strategy. We evaluated several baselines—including random, all-zero, and spectral attention filters—all of which underperformed compared to our proposed time-invariant filter method. As shown in Table 10b, ICM achieves the lowest

Table 10: (a).DeCoP demonstrates strong performance on anomaly detection benchmarks. P, R denotes precision and recall, respectively;(b).Comparison of different filter strategies across forecasting and classification tasks.

(a)

| Dataset | SMAP | | | PSM | | | MSL | | |
|---|---|---|---|---|---|---|---|---|---|
| Metrics | P | R | F1 | P | R | F1 | P | R | F1 |
| DeCoP | 87.26 | 88.36 | **87.80** | 95.69 | 94.04 | **94.86** | 84.28 | 87.55 | **85.89** |
| PatchTST | 86.80 | 85.33 | 86.06 | 96.06 | 90.73 | 93.32 | 84.54 | 86.85 | 85.68 |
| DLinear | 92.36 | 55.41 | 69.29 | 98.28 | 89.26 | 93.55 | 84.34 | 85.42 | 84.88 |
| Autoformer | 90.40 | 58.62 | 71.12 | 99.08 | 88.15 | 93.29 | 77.27 | 80.92 | 79.05 |
| Informer | 90.11 | 57.13 | 69.92 | 64.27 | 96.33 | 77.10 | 81.77 | 86.48 | 84.06 |

(b)

| Filters | ICM | | All-Zeros | | Random | | Spectral Att. | |
|---|---|---|---|---|---|---|---|---|
| Metric | MSE | MAE | MSE | MAE | MSE | MAE | MSE | MAE |
| ETTh1 | **0.401** | **0.421** | 0.406 | 0.424 | 0.403 | 0.423 | 0.404 | 0.423 |
| Metric | ACC | F1 | ACC | F1 | ACC | F1 | ACC | F1 |
| Epilepsy | **0.955** | **0.927** | 0.940 | 0.904 | 0.944 | 0.905 | 0.952 | 0.924 |

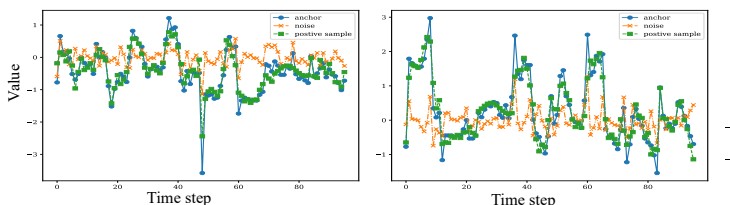

Figure 9: **The visualization of generated positive sample pairs using ICM**. These generated positive sample pairs from each forecasting dataset with a sequence length of 100. The variable index represents the relative order of channels within each dataset. The blue line indicates the original anchor sample, while the green and orange lines represent the positive sample and filtered noise, respectively. The positive sample (green) preserves the primary characteristics of the anchor sample while exhibiting controlled variations in amplitude and temporal fluctuations.

error across both forecasting and classification tasks. On the Epilepsy dataset, for example, the all-zero filter reduced F1 by 3.1% relative to ICM, demonstrating its limited efficacy. Top-K time-invariant filtering effectively generates noise-controlled positive samples, even in the early stages of training. In contrast, learnable filters often struggle to provide the stable supervision required for effective representation learning, particularly during the early stages of training. Moreover, our time-invariant filter is a parameter-free module. Contrastively, other competing strategies can introduce a performance drop while simultaneously requiring additional parameters.

# F PARAMETER SENSITIVITY ANALYSIS

## F.1 THE ROBUSTNESS OF IPN

**The analysis of parameter $\alpha_{initial}$.** We evaluate the sensitivity of IPN to the initialization of the scaling parameter $\alpha_{initial}$. As shown in Table 11, IPN consistently achieves stable performance across a wide range of $\alpha_{initial}$ values (0.01 to 0.5) on all three datasets. The observed variations in MSE and MAE are minimal, indicating that IPN is robust to the choice of $\alpha_{initial}$ and does not require careful tuning for effective performance.

Table 11: DeCoP remains stable around different $\alpha_{initial}$ on three different datasets.

| $\alpha_{initial}$ | ETTh1 | | ETTm1 | | Weather | |
|---|---|---|---|---|---|---|
| | MSE | MAE | MSE | MAE | MSE | MAE |
| 0.01 | 0.401 | 0.421 | 0.223 | 0.259 | 0.223 | 0.259 |
| 0.1 | 0.403 | 0.422 | 0.223 | 0.259 | 0.224 | 0.260 |
| 0.2 | 0.404 | 0.422 | 0.224 | 0.260 | 0.223 | 0.260 |
| 0.5 | 0.404 | 0.422 | 0.224 | 0.260 | 0.223 | 0.260 |

**The analysis of patch size $P$.** We investigate the robustness of the IPN module under varying patch lengths. Specifically, we assess the model's sensitivity to the patch size P=2, 4, 8, 12, 16, 24, 32, 40, with the look-back window fixed at 512 and the stride set equal to the patch length to avoid overlap. The forecasting horizon is fixed at 96 time steps. Figure 10 illustrates the MSE scores across three representative datasets: ETTh1, ETTm1, and Weather. The results demonstrate that the performance of the IPN module remains consistently stable for a wide range of patch lengths, particularly within the interval $P = 8$ to $P = 40$. This indicates that the IPN module is largely invariant to moderate changes in patch configuration, highlighting its robustness. Interestingly, while very small patch lengths (e.g., $P = 2$) occasionally result in slight performance degradation, larger

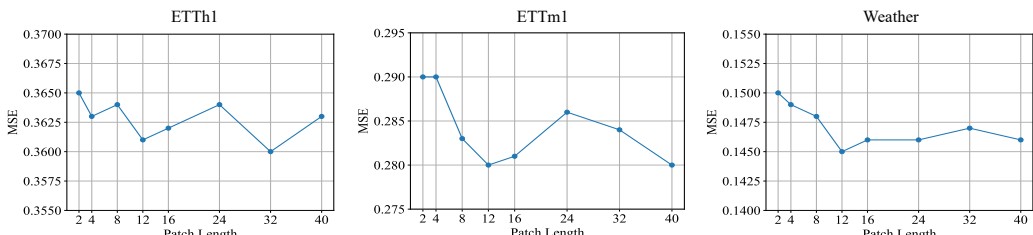

Figure 10: **IPN Module Shows Robust Performance Across Patch Sizes.** MSE scores are evaluated for patch lengths P = [2, 4, 8, 12, 16, 24, 32, 40] with a fixed look-back window of 512 and prediction length of 96. The results indicate small variation in MSE values, particularly for P = [8, 40], highlighting the IPN module's robustness.

patches consistently yield strong results. This observation suggests that using moderately large patch sizes can improve stability and performance, depending on the dataset's temporal structure. Overall, these findings confirm that the IPN module adapts effectively to different patch granularities without extensive hyperparameter tuning. This flexibility is crucial for practical deployment across datasets with diverse characteristics, reinforcing the IPN module's generalization capability.

### F.2 THE EFFECTIVENESS OF ICM

To further assess the contribution of the ICM module, we conduct a detailed evaluation of its impact under various filtering intensities on both forecasting and classification tasks. ICM is designed to filter noisy, high-frequency components in time series signals during pretraining, improving the quality of global semantics for dependency-controlled learning. By applying frequency-domain filtering, ICM allows the model to focus on temporally stable positive pairs, enhancing global representation learning at the latent level.

We investigate the effect of the $\beta$ parameter, which determines the proportion of high-amplitude frequency components retained during filtering. Higher $\beta$ values correspond to stronger filtering (i.e., more low-energy frequencies are removed). Table 12 summarizes the results on ETTh1, ETTm1, and Epilepsy datasets with $\beta \in \{0.0, 0.1, 0.2, 0.4\}$. On the forecasting tasks, we observe consistent improvements in both MSE and MAE when using non-zero $\beta$ values. For instance, on the ETTm1 dataset, the average MSE decreases from 0.350 (no filtering) to 0.342 at $\beta = 0.1$, with corresponding MAE also decreasing from 0.378 to 0.376. The performance remains relatively stable for $\beta$ values up to 0.4, suggesting that ICM is robust to a wide range of filtering intensities. Similar trends are observed on the ETTh1 dataset, where MSE improves from 0.403 to 0.401 when $\beta = 0.1$ or $\beta = 0.2$. These results confirm that mild frequency filtering enhances temporal modeling by removing distracting noise without compromising meaningful signal components. From a signal processing perspective, ICM serves as a soft spectral denoiser that targets low-amplitude components often associated with sensor drift, random fluctuations, or local outliers. By suppressing these perturbations and preserving dominant frequencies, ICM helps the model learn representations that generalize more effectively across input variations and time horizons. This benefit is especially pronounced for long-horizon forecasting (e.g., 720-step prediction), where accumulated noise tends to degrade performance more severely.

We also evaluate ICM's effectiveness on the Epilepsy classification task. Without filtering ($\beta = 0.0$), the model achieves 94.61% accuracy and 91.30% F1 score. Enabling frequency filtering with $\beta = 0.1$ increases performance to 95.4% accuracy and 92.6% F1, and the performance remains stable for higher $\beta$ values. This demonstrates that ICM not only improves regression objectives, but also preserves class-discriminative patterns while removing task-irrelevant spectral artifacts. Importantly, the $\beta$ analysis shows that ICM is not sensitive to precise hyperparameter settings; gains are observable as long as minimal filtering is applied ($\beta > 0$). This property simplifies deployment in real-world scenarios, where robust performance under moderate tuning is often preferred. Moreover, since ICM is only used during pretraining, it imposes no additional inference cost. In summary, ICM complements the hierarchical modeling of dependencies in DCL by enhancing global repre-

Table 12: Impact of $\beta$ filtering on forecasting and classification performance across ETTh1, ETTm1, and Epilepsy datasets. For forecasting, MSE/MAE are reported. For classification, Accuracy (ACC) and F1 score are presented.

| Dataset | $\beta$ | 0 | | 0.1 | | 0.2 | | 0.4 | |
|---|---|---|---|---|---|---|---|---|---|
| | Metric | MSE | MAE | MSE | MAE | MSE | MAE | MSE | MAE |
| **ETTh1** | 96 | 0.362 | 0.391 | 0.360 | 0.390 | 0.360 | 0.390 | 0.361 | 0.390 |
| | 192 | 0.395 | 0.411 | 0.394 | 0.410 | 0.394 | 0.410 | 0.394 | 0.411 |
| | 336 | 0.418 | 0.427 | 0.417 | 0.427 | 0.417 | 0.427 | 0.418 | 0.427 |
| | 720 | 0.439 | 0.460 | 0.433 | 0.455 | 0.433 | 0.455 | 0.439 | 0.459 |
| | Avg | 0.403 | 0.422 | 0.401 | 0.421 | 0.401 | 0.421 | 0.403 | 0.422 |
| | Metric | MSE | MAE | MSE | MAE | MSE | MAE | MSE | MAE |
| **ETTm1** | 96 | 0.286 | 0.342 | 0.281 | 0.340 | 0.284 | 0.342 | 0.284 | 0.341 |
| | 192 | 0.331 | 0.368 | 0.325 | 0.366 | 0.329 | 0.367 | 0.329 | 0.367 |
| | 336 | 0.364 | 0.386 | 0.353 | 0.387 | 0.356 | 0.387 | 0.361 | 0.384 |
| | 720 | 0.421 | 0.418 | 0.410 | 0.413 | 0.416 | 0.415 | 0.416 | 0.418 |
| | Avg | 0.350 | 0.378 | 0.342 | 0.376 | 0.346 | 0.378 | 0.347 | 0.377 |
| **Epilepsy** | Metric | ACC | F1 | ACC | F1 | ACC | F1 | ACC | F1 |
| | Value | 94.61 | 91.30 | 95.4 | 92.6 | 95.1 | 92.2 | 95.1 | 92.2 |

sentation learning, and plays a critical role in enabling DeCoP to operate reliably under multiscale and non-stationary conditions.

### F.3 THE GENERALIZABILITY OF DCL STRATEGY

The DCL method is designed to model hierarchical temporal dependencies by aggregating information across multiple temporal resolutions. To evaluate the generalization ability of DCL across diverse input scales, we conduct experiments by varying the look-back length $L \in \{96, 192, 336, 512, 720\}$ while keeping the patch size and stride fixed. This setting isolates the impact of input sequence length while holding the architectural capacity constant. We report forecasting performance compare to CycleNet, a state-of-the-art baseline that adopts a linear or MLP-based temporal backbone, on three representative datasets (ETTh1, ETTm1, and Weather) in Table 13.

Across all datasets and input lengths, DeCoP consistently outperforms CycleNet, demonstrating its robustness to varying temporal contexts. For instance, on ETTm1, DeCoP achieves the lowest average MSE of 0.341 and MAE of 0.376, with particularly strong results at $L = 192$ and $L = 336$, indicating its capacity to adaptively extract meaningful patterns at medium-range scales. On ETTh1, DeCoP shows clear advantages at longer horizons (e.g., $L = 512, 720$), outperforming CycleNet in both short- and long-term forecasting regimes. On the Weather dataset, DeCoP outperforms CycleNet by a significant margin across all settings, especially at $L = 720$ where it achieves MSE = 0.222 and MAE = 0.260, confirming its ability to handle highly periodic or irregular signals with long-range dependencies. These results confirm that DCL not only generalizes across datasets with varying dynamics but also scales well with increasing input length without degrading performance. In contrast, CycleNet's performance tends to deteriorate or plateau under long input sequences, which we attribute to its static modeling capacity and lack of explicit multi-scale mechanisms. By progressively expanding the receptive field through hierarchical windows, DCL adaptively captures dependencies at different temporal scopes and mitigates issues such as temporal overfitting. In summary, DCL provides robustness to both local and global temporal changes, making it an effective backbone for time series modeling under real-world scenarios with varying historical contexts.

### F.4 ABLATION STUDY ON THE CONTRASTIVE LOSS WEIGHT $\gamma$

We investigate the sensitivity of model performance to the contrastive loss weight $\gamma$, which controls the contribution of $\mathcal{L}_{cl}$ during training. As shown in Table 14, smaller values of $\gamma$ (e.g., 0.1) yield competitive results on regression tasks such as ETTh2 and Weather, with minimal variation across settings. In contrast, larger values of $\gamma$ significantly improve performance on high-level classification tasks, as evidenced by a steady increase in

Table 13: **Dependence-Controlled Learning Effectiveness Across Different Look-Back Times.** This table compares the performance of DeCoP and CycleNet across various look-back windows and prediction horizons on three different datasets. The best result for each dataset is marked in yellow, and the best under each look-back time is marked in pink. DeCoP achieve allover best in each dataset and can get best result under the same look-back time once the input length larger than 96.

| Dataset | Models | Look-back time | 96 | | 192 | | 336 | | 512 | | 720 | |
|---|---|---|---|---|---|---|---|---|---|---|---|---|
| | | | MSE | MAE | MSE | MAE | MSE | MAE | MSE | MAE | MSE | MAE |
| ETTh1 | DeCoP | 96 | 0.384 | 0.394 | 0.380 | 0.396 | 0.370 | 0.393 | 0.361 | 0.390 | 0.363 | 0.394 |
| | | 192 | 0.436 | 0.424 | 0.421 | 0.418 | 0.403 | 0.413 | 0.394 | 0.410 | 0.400 | 0.418 |
| | | 336 | 0.478 | 0.446 | 0.453 | 0.436 | 0.428 | 0.429 | 0.414 | 0.427 | 0.426 | 0.436 |
| | | 720 | 0.478 | 0.470 | 0.451 | 0.460 | 0.442 | 0.460 | 0.437 | 0.458 | 0.453 | 0.472 |
| | | Avg | 0.444 | 0.433 | 0.426 | 0.428 | 0.411 | 0.424 | **0.401** | **0.421** | 0.411 | 0.430 |
| | CycleNet | 96 | 0.378 | 0.391 | - | - | 0.374 | 0.396 | - | - | 0.379 | 0.403 |
| | | 192 | 0.426 | 0.419 | - | - | 0.406 | 0.415 | - | - | 0.416 | 0.425 |
| | | 336 | 0.464 | 0.439 | - | - | 0.431 | 0.43 | - | - | 0.447 | 0.445 |
| | | 720 | 0.461 | 0.46 | - | - | 0.45 | 0.464 | - | - | 0.477 | 0.483 |
| | | Avg | 0.432 | 0.427 | - | - | 0.415 | 0.426 | - | - | 0.43 | 0.439 |
| ETTm1 | DeCoP | 96 | 0.314 | 0.356 | 0.288 | 0.340 | 0.282 | 0.338 | 0.280 | 0.338 | 0.296 | 0.348 |
| | | 192 | 0.358 | 0.380 | 0.323 | 0.363 | 0.324 | 0.365 | 0.323 | 0.365 | 0.332 | 0.369 |
| | | 336 | 0.387 | 0.400 | 0.357 | 0.386 | 0.355 | 0.385 | 0.353 | 0.385 | 0.364 | 0.386 |
| | | 720 | 0.449 | 0.435 | 0.417 | 0.421 | 0.411 | 0.418 | 0.409 | 0.418 | 0.412 | 0.415 |
| | | Avg | 0.377 | 0.393 | 0.346 | 0.377 | 0.343 | 0.377 | **0.342** | **0.376** | 0.351 | 0.379 |
| | CycleNet | 96 | 0.319 | 0.36 | - | - | 0.299 | 0.348 | - | - | 0.307 | 0.353 |
| | | 192 | 0.36 | 0.381 | - | - | 0.334 | 0.367 | - | - | 0.337 | 0.371 |
| | | 336 | 0.389 | 0.403 | - | - | 0.368 | 0.386 | - | - | 0.364 | 0.387 |
| | | 720 | 0.447 | 0.441 | - | - | 0.417 | 0.414 | - | - | 0.41 | 0.411 |
| | | Avg | 0.379 | 0.396 | - | - | 0.355 | 0.379 | - | - | 0.355 | 0.381 |
| Weather | DeCoP | 96 | 0.175 | 0.216 | 0.157 | 0.201 | 0.148 | 0.195 | 0.145 | 0.193 | 0.144 | 0.192 |
| | | 192 | 0.222 | 0.256 | 0.201 | 0.242 | 0.192 | 0.238 | 0.190 | 0.237 | 0.189 | 0.236 |
| | | 336 | 0.277 | 0.296 | 0.255 | 0.283 | 0.244 | 0.278 | 0.242 | 0.278 | 0.242 | 0.279 |
| | | 720 | 0.352 | 0.346 | 0.332 | 0.336 | 0.318 | 0.331 | 0.314 | 0.329 | 0.313 | 0.331 |
| | | Avg | 0.256 | 0.278 | 0.236 | 0.266 | 0.226 | 0.261 | **0.223** | **0.259** | 0.222 | 0.260 |
| | CycleNet | 96 | 0.158 | 0.203 | - | - | 0.148 | 0.2 | - | - | 0.149 | 0.203 |
| | | 192 | 0.207 | 0.247 | - | - | 0.19 | 0.24 | - | - | 0.192 | 0.244 |
| | | 336 | 0.262 | 0.289 | - | - | 0.243 | 0.283 | - | - | 0.242 | 0.283 |
| | | 720 | 0.344 | 0.344 | - | - | 0.322 | 0.339 | - | - | 0.312 | 0.333 |
| | | Avg | 0.243 | 0.271 | - | - | 0.226 | 0.266 | - | - | 0.224 | 0.266 |

accuracy and F1 score on the Epilepsy dataset. These results suggest that contrastive supervision is especially beneficial for learning global representations in classification settings.

**The visualization of time-invariant filter.** We visualize the effect of our time-invariant filtering on both forecasting (ETTh1) and classification (SleepEEG) tasks. As shown in Figure 11(a), the ICM successfully removes noise while generating positive samples that preserve the temporal structure of the anchor. The filtered noise (orange) exhibits near-zero mean and low variance, resembling white noise, which aligns with the central limit theorem. Figures 11(b) and 11(c) further analyze the retained and discarded frequencies. The green dots denote time-invariant frequencies preserved by ICM, while orange dots indicate time-variant components filtered out. Notably, despite their lower amplitude, the retained green frequencies remain consistent across both datasets, validating their stability and importance for learning generalizable patterns. This demonstrates that the ICM mechanism prioritizes informative, time-invariant signals over low-amplitude and unstable variations, preventing information loss and providing stable positive pairs during pretraining.

Table 14: DeCoP remains stable performance around different $\gamma$ for ETTh2 and Weather datasets.

| $\gamma$ | ETTh2 | | Weather | | Epilepsy | |
|---|---|---|---|---|---|---|
| | MSE | MAE | MSE | MAE | Accuracy | F1 |
| 0.01 | 0.336 | 0.385 | 0.224 | 0.261 | 93.38 | 89.41 |
| 0.1 | 0.336 | 0.385 | 0.224 | 0.261 | 93.84 | 90.08 |
| 0.3 | 0.335 | 0.385 | 0.224 | 0.261 | 94.86 | 91.26 |
| 0.5 | 0.336 | 0.385 | 0.225 | 0.261 | 95.53 | 92.86 |

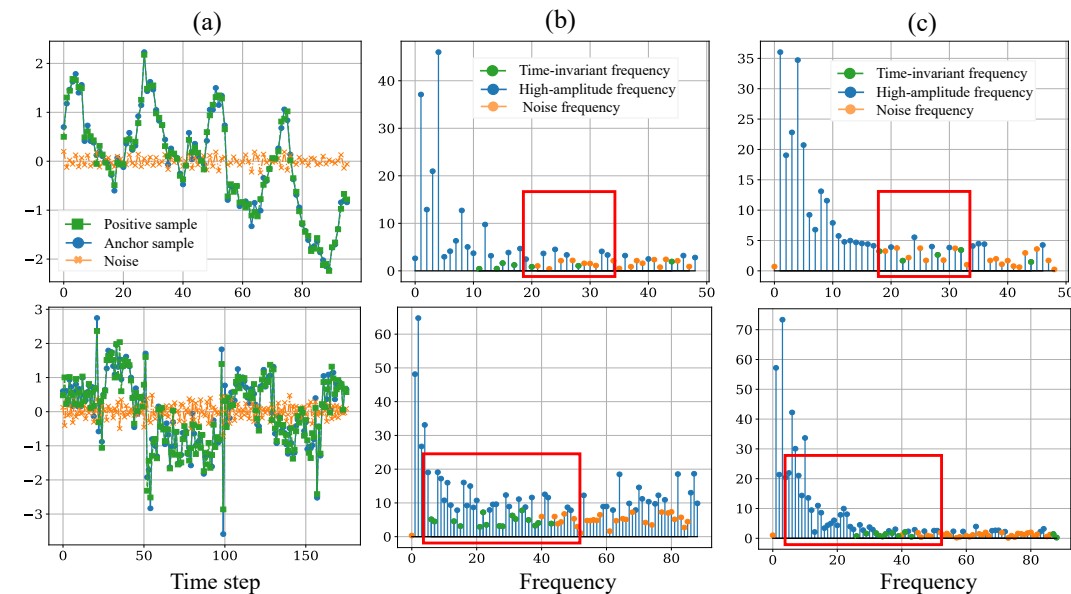

Figure 11: The ICM filters time-variant noise while preserving meaningful time-invariant frequencies. The top and bottom rows show this process on examples from two different tasks: the ETTh1 forecasting dataset (sequence length 100) and the SleepEEG classification dataset (sequence length 178), respectively. Column (a) Visualization of the anchor sample, filtered noise, and generated positive sample on the last channel of the ETTh1 and SleepEEG dataset. The filtered noise (orange) resembles white noise with zero mean, consistent with the central limit theorem. Column (b) Amplitude spectrum of (a), where green dots denote time-invariant frequencies retained by ICM, and orange dots indicate time-variant frequencies removed by the $Fmask$. Column (c) Although the green-highlighted frequencies in (b) show lower amplitude, they remain prominent across (c), confirming their time-invariant nature and importance for capturing stable patterns.

# G    FULL ABLATION STUDY RESULTS

To thoroughly assess the contribution of each module in our proposed DeCoP framework, we conduct comprehensive ablation studies on both forecasting and classification tasks, across both in-domain and cross-domain settings.

## G.1    ABLATION STUDY ON FORECASTING TASKS

These studies evaluate the necessity and effectiveness of the key components: Instance-wise Patch Normalization (IPN), the Instance-level Contrastive Module (ICM), and the Dependency-Controlled Learning (DCL) mechanism. We summarize forecasting ablation results in Table 15 and Table 16, and classification results in Table 17.

We design four ablation variants: (1) *w/o IPN*, (2) *w/o ICM*, (3) *PI* (fully Patch-Independent DCL), and (4) *PD* (fully Patch-Dependent DCL). These are compared against the full DeCoP model across six datasets for forecasting (ETTh1, ETTh2, ETTm1, ETTm2, Weather, Electricity) and three for classification (Epilepsy, FD-B, EMG).

Across all datasets and forecast horizons, the full DeCoP model consistently outperforms its ablated variants. In in-domain forecasting as shown in Table 15, DeCoP achieves the lowest average MSE and MAE in nearly every case. For example, on ETTh2, DeCoP obtains an average MSE of 0.333, significantly outperforming *w/o IPN* (0.337), *w/o ICM* (0.385), and both PI and PD variants (0.335 and 0.383, respectively). The impact of the IPN and ICM is particularly evident on datasets like ETTm1 and Electricity, where the removal of either leads to a substantial drop in performance—highlighting their critical role in ICM and frequency-based noise suppression.

Table 15: Full ablation studies ablation studies were conducted on in-domain learning tasks. The experiments focus on forecasting future time points $F \in \{96, 192, 336, 720\}$ based on a look-back window of 512 past time points. Best results are denoted by **bold**.

| Scenarios | Models Metric | w/o IPN | | w/o ICM | | PI | | PD | | **DeCoP** | |
|---|---|---|---|---|---|---|---|---|---|---|---|
| | | MSE | MAE | MSE | MAE | MSE | MAE | MSE | MAE | MSE | MAE |
| ETTh1→ETTh1 | 96 | 0.365 | 0.392 | 0.362 | 0.391 | 0.364 | 0.392 | 0.363 | 0.391 | 0.360 | 0.390 |
| | 192 | 0.400 | 0.413 | 0.395 | 0.411 | 0.400 | 0.415 | 0.398 | 0.413 | 0.394 | 0.410 |
| | 336 | 0.423 | 0.429 | 0.418 | 0.427 | 0.416 | 0.426 | 0.419 | 0.426 | 0.417 | 0.427 |
| | 720 | 0.434 | 0.455 | 0.439 | 0.460 | 0.444 | 0.462 | 0.442 | 0.460 | 0.433 | 0.455 |
| | Average | 0.406 | 0.422 | 0.403 | 0.422 | 0.406 | 0.424 | 0.405 | 0.423 | **0.401** | **0.421** |
| ETTh2→ETTh2 | 96 | 0.275 | 0.336 | 0.273 | 0.333 | 0.272 | 0.334 | 0.267 | 0.332 | 0.267 | 0.332 |
| | 192 | 0.331 | 0.375 | 0.330 | 0.374 | 0.329 | 0.373 | 0.330 | 0.374 | 0.328 | 0.373 |
| | 336 | 0.358 | 0.400 | 0.360 | 0.398 | 0.355 | 0.398 | 0.359 | 0.399 | 0.353 | 0.397 |
| | 720 | 0.386 | 0.429 | 0.385 | 0.427 | 0.385 | 0.427 | 0.383 | 0.427 | 0.382 | 0.425 |
| | Average | 0.337 | 0.385 | 0.337 | 0.383 | 0.335 | 0.383 | 0.335 | 0.383 | **0.333** | **0.382** |
| ETTm1→ETTm1 | 96 | 0.290 | 0.345 | 0.286 | 0.342 | 0.292 | 0.349 | 0.290 | 0.345 | 0.281 | 0.340 |
| | 192 | 0.330 | 0.370 | 0.331 | 0.368 | 0.330 | 0.373 | 0.323 | 0.364 | 0.325 | 0.366 |
| | 336 | 0.366 | 0.386 | 0.364 | 0.386 | 0.364 | 0.392 | 0.356 | 0.386 | 0.353 | 0.387 |
| | 720 | 0.424 | 0.416 | 0.421 | 0.418 | 0.405 | 0.416 | 0.411 | 0.413 | 0.410 | 0.413 |
| | Average | 0.352 | 0.379 | 0.350 | 0.378 | 0.348 | 0.382 | 0.345 | 0.377 | **0.342** | **0.376** |
| ETTm2→ETTm2 | 96 | 0.164 | 0.255 | 0.164 | 0.256 | 0.163 | 0.253 | 0.167 | 0.252 | 0.163 | 0.255 |
| | 192 | 0.220 | 0.296 | 0.221 | 0.294 | 0.216 | 0.289 | 0.220 | 0.295 | 0.217 | 0.290 |
| | 336 | 0.272 | 0.330 | 0.273 | 0.331 | 0.269 | 0.323 | 0.268 | 0.327 | 0.266 | 0.324 |
| | 720 | 0.356 | 0.382 | 0.358 | 0.384 | 0.393 | 0.408 | 0.353 | 0.381 | 0.350 | 0.377 |
| | Average | 0.253 | 0.316 | 0.254 | 0.316 | 0.260 | 0.318 | 0.252 | 0.314 | **0.249** | **0.311** |
| Weather→Weather | 96 | 0.151 | 0.201 | 0.150 | 0.199 | 0.156 | 0.206 | 0.146 | 0.197 | 0.146 | 0.193 |
| | 192 | 0.194 | 0.240 | 0.193 | 0.239 | 0.196 | 0.241 | 0.193 | 0.242 | 0.190 | 0.236 |
| | 336 | 0.244 | 0.279 | 0.243 | 0.278 | 0.247 | 0.282 | 0.245 | 0.281 | 0.242 | 0.277 |
| | 720 | 0.316 | 0.331 | 0.317 | 0.332 | 0.317 | 0.334 | 0.329 | 0.339 | 0.315 | 0.330 |
| | Average | 0.226 | 0.263 | 0.226 | 0.262 | 0.229 | 0.266 | 0.228 | 0.265 | **0.223** | **0.259** |
| Electricity→Electricity | 96 | 0.132 | 0.227 | 0.132 | 0.228 | 0.135 | 0.230 | 0.130 | 0.226 | 0.127 | 0.222 |
| | 192 | 0.149 | 0.243 | 0.149 | 0.243 | 0.150 | 0.244 | 0.148 | 0.242 | 0.145 | 0.239 |
| | 336 | 0.165 | 0.259 | 0.165 | 0.260 | 0.166 | 0.260 | 0.164 | 0.259 | 0.161 | 0.257 |
| | 720 | 0.204 | 0.293 | 0.204 | 0.293 | 0.205 | 0.293 | 0.203 | 0.293 | 0.195 | 0.289 |
| | Average | 0.162 | 0.255 | 0.163 | 0.256 | 0.164 | 0.257 | 0.162 | 0.255 | **0.157** | **0.251** |

Moreover, the ablation of the DCL mechanism provides further insights into the importance of dependency control. Among the two variants, the Patch-Independent (PI) version consistently underperforms, suggesting that treating patches as completely independent fails to capture important hierarchical dependencies. The PD variant (fully dependent) performs slightly better, but still falls short of DeCoP, indicating that overly strong dependency assumptions may lead to overfitting or loss of flexibility. The superior performance of DeCoP, which adopts a partially dependent design, confirms that adaptive dependency modeling offers the best trade-off between representation expressiveness and regularization.

In the cross-domain forecasting setup (Table 16), where models are transferred across datasets, the performance gaps become even more pronounced. For instance, on the ETTm2 → ETTh1 transfer task, DeCoP achieves an average MSE of 0.404, while all ablation baselines perform worse, with the PI variant dropping to 0.428 and the ICM-ablated version reaching 0.412. This widening performance gap under domain shift further emphasizes the importance of ICM and DCL in enabling generalizable temporal representations. Additionally, results on Weather → ETTh1 demonstrate that DeCoP can effectively transfer temporal priors from one domain to another, outperforming baselines even under significant distributional changes.

### G.2 ABLATION STUDY ON CLASSIFICATION TASKS

For classification tasks as shown in Table 17, DeCoP similarly outperforms all ablation variants in both in-domain and cross-domain settings. In the in-domain Epilepsy classification task, DeCoP attains an average score of 94.20%, compared to 92.19% and 92.77% for the *w/o IPN* and *w/o ICM* variants, and substantially higher than PI (90.26%) and PD (71.69%). In the more challeng-

Table 16: Full ablation studies ablation studies were conducted on cross-domain transfer tasks to ETTh1 and ETTm1 datasets. The experiments focus on forecasting future time points $F \in \{96, 192, 336, 720\}$ based on a look-back window of 512 past time points. Best results are denoted by **bold**.

| Scenarios | Models Metric | w/o IPN | | w/o ICM | | PI | | PD | | **DeCoP** | |
|---|---|---|---|---|---|---|---|---|---|---|---|
| | | MSE | MAE | MSE | MAE | MSE | MAE | MSE | MAE | MSE | MAE |
| ETTh2 ↓ ETTh1 | 96 | 0.364 | 0.392 | 0.364 | 0.392 | 0.364 | 0.391 | 0.365 | 0.391 | 0.362 | 0.391 |
| | 192 | 0.396 | 0.411 | 0.396 | 0.412 | 0.401 | 0.414 | 0.401 | 0.414 | 0.394 | 0.411 |
| | 336 | 0.421 | 0.429 | 0.428 | 0.428 | 0.424 | 0.431 | 0.425 | 0.431 | 0.417 | 0.427 |
| | 720 | 0.446 | 0.462 | 0.442 | 0.460 | 0.443 | 0.460 | 0.443 | 0.463 | 0.438 | 0.459 |
| | Average | 0.407 | 0.423 | 0.407 | 0.423 | 0.408 | 0.424 | 0.408 | 0.425 | **0.403** | **0.422** |
| ETTm1 ↓ ETTh1 | 96 | 0.363 | 0.391 | 0.366 | 0.394 | 0.363 | 0.390 | 0.364 | 0.390 | 0.364 | 0.392 |
| | 192 | 0.395 | 0.411 | 0.398 | 0.415 | 0.396 | 0.411 | 0.400 | 0.414 | 0.395 | 0.411 |
| | 336 | 0.422 | 0.432 | 0.421 | 0.429 | 0.422 | 0.429 | 0.423 | 0.430 | 0.419 | 0.430 |
| | 720 | 0.451 | 0.468 | 0.445 | 0.464 | 0.442 | 0.460 | 0.443 | 0.464 | 0.441 | 0.459 |
| | Average | 0.408 | 0.426 | 0.407 | 0.425 | 0.406 | 0.422 | 0.407 | 0.425 | **0.405** | **0.423** |
| ETTm2 ↓ ETTh1 | 96 | 0.363 | 0.391 | 0.371 | 0.398 | 0.364 | 0.391 | 0.364 | 0.389 | 0.363 | 0.392 |
| | 192 | 0.396 | 0.411 | 0.402 | 0.417 | 0.397 | 0.411 | 0.401 | 0.414 | 0.395 | 0.411 |
| | 336 | 0.421 | 0.427 | 0.426 | 0.432 | 0.429 | 0.428 | 0.424 | 0.430 | 0.418 | 0.429 |
| | 720 | 0.442 | 0.459 | 0.449 | 0.465 | 0.444 | 0.460 | 0.442 | 0.459 | 0.441 | 0.461 |
| | Average | 0.406 | 0.422 | 0.412 | 0.428 | 0.408 | 0.423 | 0.408 | 0.423 | **0.404** | **0.423** |
| Weather ↓ ETTh1 | 96 | 0.363 | 0.392 | 0.365 | 0.393 | 0.364 | 0.391 | 0.365 | 0.391 | 0.365 | 0.392 |
| | 192 | 0.396 | 0.413 | 0.397 | 0.413 | 0.397 | 0.411 | 0.398 | 0.412 | 0.397 | 0.412 |
| | 336 | 0.422 | 0.430 | 0.421 | 0.428 | 0.422 | 0.428 | 0.424 | 0.431 | 0.421 | 0.428 |
| | 720 | 0.447 | 0.467 | 0.442 | 0.461 | 0.443 | 0.460 | 0.442 | 0.459 | 0.439 | 0.458 |
| | Average | 0.407 | 0.425 | 0.406 | 0.424 | 0.407 | 0.423 | 0.407 | 0.423 | **0.405** | **0.422** |
| ETTh2 ↓ ETTm1 | 96 | 0.287 | 0.343 | 0.286 | 0.342 | 0.297 | 0.350 | 0.285 | 0.341 | 0.282 | 0.340 |
| | 192 | 0.328 | 0.368 | 0.328 | 0.368 | 0.341 | 0.372 | 0.327 | 0.366 | 0.323 | 0.365 |
| | 336 | 0.357 | 0.386 | 0.356 | 0.386 | 0.359 | 0.389 | 0.360 | 0.386 | 0.359 | 0.389 |
| | 720 | 0.417 | 0.422 | 0.421 | 0.424 | 0.418 | 0.419 | 0.414 | 0.415 | 0.408 | 0.413 |
| | Average | 0.347 | 0.380 | 0.348 | 0.380 | 0.354 | 0.382 | 0.346 | 0.377 | **0.343** | **0.377** |
| ETTh1 ↓ ETTm1 | 96 | 0.287 | 0.343 | 0.287 | 0.343 | 0.295 | 0.349 | 0.288 | 0.345 | 0.283 | 0.340 |
| | 192 | 0.329 | 0.369 | 0.328 | 0.368 | 0.334 | 0.374 | 0.329 | 0.366 | 0.329 | 0.367 |
| | 336 | 0.356 | 0.387 | 0.357 | 0.387 | 0.366 | 0.390 | 0.364 | 0.387 | 0.357 | 0.389 |
| | 720 | 0.426 | 0.424 | 0.420 | 0.424 | 0.410 | 0.417 | 0.420 | 0.419 | 0.414 | 0.417 |
| | Average | 0.350 | 0.381 | 0.348 | 0.380 | 0.351 | 0.382 | 0.350 | 0.379 | **0.346** | **0.379** |
| ETTm2 ↓ ETTm1 | 96 | 0.287 | 0.342 | 0.286 | 0.342 | 0.297 | 0.350 | 0.282 | 0.340 | 0.283 | 0.342 |
| | 192 | 0.330 | 0.368 | 0.330 | 0.368 | 0.333 | 0.371 | 0.324 | 0.365 | 0.323 | 0.364 |
| | 336 | 0.356 | 0.387 | 0.365 | 0.386 | 0.363 | 0.389 | 0.359 | 0.386 | 0.355 | 0.385 |
| | 720 | 0.418 | 0.422 | 0.420 | 0.422 | 0.408 | 0.415 | 0.413 | 0.419 | 0.406 | 0.414 |
| | Average | 0.348 | 0.380 | 0.350 | 0.379 | 0.350 | 0.381 | 0.344 | 0.377 | **0.342** | **0.376** |
| Weather ↓ ETTm1 | 96 | 0.290 | 0.345 | 0.286 | 0.343 | 0.291 | 0.347 | 0.291 | 0.344 | 0.284 | 0.341 |
| | 192 | 0.331 | 0.368 | 0.332 | 0.371 | 0.327 | 0.371 | 0.328 | 0.369 | 0.325 | 0.365 |
| | 336 | 0.364 | 0.385 | 0.364 | 0.386 | 0.364 | 0.392 | 0.362 | 0.386 | 0.357 | 0.384 |
| | 720 | 0.419 | 0.421 | 0.418 | 0.422 | 0.411 | 0.417 | 0.417 | 0.418 | 0.417 | 0.414 |
| | Average | 0.351 | 0.380 | 0.350 | 0.380 | 0.348 | 0.382 | 0.349 | 0.379 | **0.345** | **0.376** |

ing cross-domain scenarios, such as SleepEEG → FD-B and EMG, DeCoP maintains top perfor­mance—achieving 93.97% and 100% average scores, respectively—while all ablated variants suffer from severe accuracy drops, especially when dependency modeling is removed. Additionally, we include an additional classification setting using the SleepEEG → Gesture dataset in this section, and the results are consistent with those observed on other datasets.

Together, these results validate the contribution of each module in our framework. The ICM plays a vital role in filtering irrelevant frequency components while preserving semantically meaningful pat­terns, which enhances global representation quality for downstream modeling. The IPN mechanism stabilizes patch-level inputs by eliminating scale variance across instances. Finally, the DCL strat-

Table 17: Ablation study conducted on classification tasks in both in-domain and cross-domain settings. For the in-domain setting, the model is pre-trained and fine-tuned on the same dataset (Epilepsy). In the cross-domain setting, the model is pre-trained on the SleepEEG dataset and subsequently fine-tuned on various target datasets, including Epilepsy, FD-B, and EMG. AVG denotes the average of accuracy and F1 score. Best results are denoted by **bold**.

| | Scenarios | Models | Acc (%) | P (%) | R (%) | F1 (%) | Avg (%) |
|---|---|---|---|---|---|---|---|
| **In-Domain** | Epilepsy ↓ Epilepsy | w/o IPN | 94.19 | 93.64 | 87.57 | 90.19 | 92.19 |
| | | w/o ICM | 94.67 | 95.20 | 87.73 | 90.87 | 92.77 |
| | | PI | 93.02 | 94.51 | 83.27 | 87.50 | 90.26 |
| | | PD | 83.45 | 89.07 | 58.52 | 59.92 | 71.69 |
| | | **DeCoP** | 95.53 | 93.51 | 92.25 | 92.86 | **94.20** |
| **Cross-Domain** | SleepEEG ↓ Epilepsy | w/o IPN | 94.40 | 93.89 | 88.01 | 90.57 | 92.49 |
| | | w/o ICM | 94.15 | 94.20 | 86.94 | 89.99 | 92.07 |
| | | PI | 94.73 | 93.14 | 89.85 | 91.37 | 93.05 |
| | | PD | 90.44 | 93.85 | 76.21 | 81.46 | 85.95 |
| | | **DeCoP** | 95.82 | 94.23 | 92.41 | 93.28 | **94.55** |
| | SleepEEG ↓ FD-B | w/o IPN | 86.05 | 89.85 | 89.78 | 89.78 | 87.91 |
| | | w/o ICM | 82.12 | 86.93 | 86.84 | 86.77 | 84.44 |
| | | PI | 71.58 | 79.74 | 79.19 | 78.97 | 75.28 |
| | | PD | 70.51 | 78.54 | 78.26 | 78.29 | 74.40 |
| | | **DeCoP** | 93.04 | 94.92 | 94.90 | 94.90 | **93.97** |
| | SleepEEG ↓ Gesture | w/o IPN | 78.33 | 80.44 | 78.33 | 76.33 | 77.33 |
| | | w/o ICM | 80.00 | 80.21 | 80.00 | 77.83 | 78.91 |
| | | PI | 70.83 | 68.78 | 70.83 | 68.75 | 69.79 |
| | | PD | 79.17 | 77.38 | 79.17 | 77.56 | 78.36 |
| | | **DeCoP** | 81.67 | 80.99 | 81.67 | 80.10 | **80.89** |
| | SleepEEG ↓ EMG | w/o IPN | 97.56 | 94.44 | 98.04 | 95.96 | 96.76 |
| | | w/o ICM | 92.68 | 87.96 | 84.71 | 86.03 | 89.36 |
| | | PI | 87.80 | 59.09 | 66.67 | 62.39 | 75.10 |
| | | PD | 82.93 | 59.12 | 63.16 | 59.12 | 71.03 |
| | | **DeCoP** | 100.00 | 100.00 | 100.00 | 100.00 | **100.00** |

egy—particularly its partially dependent variant—proves crucial for capturing structured temporal dependencies without overfitting.

Overall, the ablation study provides empirical evidence that the combination of each module is key to DeCoP's success. These components jointly enable DeCoP to generalize across a wide range of datasets, input lengths, and tasks, delivering robust performance even in cross-domain scenarios where generalization is most challenging.

# H    FULL BENCHMARK OF TIME SERIES FORECASTING

## H.1    FULL IN-DOMAIN FORECASTING RESULTS

Table 18 reports the complete results on six benchmark datasets under the long-term forecasting setting, where the models are trained on the past 512 time points and evaluated on future horizons of 96, 192, 336, and 720 steps. Our proposed DeCoP variants, particularly $DeCoP_{MLP}$, consistently outperform existing baselines across a wide range of datasets and forecast lengths. $DeCoP_{MLP}$ achieves the best average performance in both MSE and MAE metrics.

Notably, $DeCoP_{MLP}$ sets new state-of-the-art results on ETTm2 and Weather, two challenging datasets characterized by complex seasonality and high variability. For instance, on ETTm2, $DeCoP_{MLP}$ obtains an MSE of 0.249 and MAE of 0.311, outperforming the second-best method by a substantial margin. Additionally, we observe that $DeCoP_{Linear}$—a lightweight variant of our model—already surpasses most baselines, demonstrating the strong generalization capability and robustness of the decomposition-based design even without complex nonlinear transformations.

Overall, the empirical results validate the effectiveness of DeCoP in capturing long-term temporal dependencies and mitigating error accumulation across diverse domains. The performance gains are

Table 18: Complete results of long-term forecasting tasks for the in-domain setting of forecasting the future $F \in \{96, 192, 336, 720\}$ time points based on the past 512 time points. The best results are denoted by **bold**.

| Models | | DeCoP$_{Linear}$ | | DeCoP$_{MLP}$ | | SIMMTM | | PatchTST | | CycleNet | | TimeMixer | | Dlinear | | iTransformer | | Fedformer | | Autoformer | | Informer | |
|---|---|---|---|---|---|---|---|---|---|---|---|---|---|---|---|---|---|---|---|---|---|---|---|
| Metric | | MSE | MAE | MSE | MAE | MSE | MAE | MSE | MAE | MSE | MAE | MSE | MAE | MSE | MAE | MSE | MAE | MSE | MAE | MSE | MAE | MSE | MAE |
| ETTh1 | 96 | 0.360 | 0.390 | 0.363 | 0.392 | 0.367 | 0.402 | 0.366 | 0.397 | 0.383 | 0.408 | 0.378 | 0.406 | 0.375 | 0.399 | 0.400 | 0.425 | 0.376 | 0.415 | 0.435 | 0.446 | 0.941 | 0.769 |
| | 192 | 0.394 | 0.410 | 0.395 | 0.412 | 0.401 | 0.425 | 0.431 | 0.443 | 0.410 | 0.426 | 0.444 | 0.448 | 0.405 | 0.426 | 0.427 | 0.443 | 0.423 | 0.446 | 0.456 | 0.457 | 1.007 | 0.786 |
| | 336 | 0.417 | 0.427 | 0.418 | 0.430 | 0.415 | 0.430 | 0.45 | 0.456 | 0.440 | 0.444 | 0.424 | 0.444 | 0.439 | 0.443 | 0.457 | 0.465 | 0.444 | 0.462 | 0.486 | 0.487 | 1.038 | 0.784 |
| | 720 | 0.433 | 0.455 | 0.451 | 0.463 | 0.455 | 0.464 | 0.472 | 0.484 | 0.487 | 0.482 | 0.485 | 0.485 | 0.472 | 0.490 | 0.631 | 0.574 | 0.469 | 0.492 | 0.515 | 0.517 | 1.144 | 0.857 |
| | Avg | **0.401** | **0.421** | 0.408 | 0.424 | 0.404 | 0.428 | 0.43 | 0.445 | 0.430 | 0.440 | 0.432 | 0.446 | 0.423 | 0.440 | 0.479 | 0.477 | 0.428 | 0.454 | 0.473 | 0.477 | 1.033 | 0.799 |
| ETTh2 | 96 | 0.268 | 0.333 | 0.271 | 0.334 | 0.288 | 0.347 | 0.284 | 0.343 | 0.299 | 0.355 | 0.280 | 0.352 | 0.289 | 0.353 | 0.299 | 0.359 | 0.332 | 0.374 | 0.332 | 0.368 | 1.549 | 0.952 |
| | 192 | 0.328 | 0.373 | 0.334 | 0.377 | 0.346 | 0.385 | 0.355 | 0.387 | 0.354 | 0.394 | 0.367 | 0.400 | 0.383 | 0.418 | 0.377 | 0.406 | 0.407 | 0.446 | 0.426 | 0.434 | 3.792 | 1.542 |
| | 336 | 0.353 | 0.397 | 0.360 | 0.404 | 0.363 | 0.401 | 0.379 | 0.411 | 0.392 | 0.425 | 0.385 | 0.420 | 0.448 | 0.465 | 0.429 | 0.442 | 0.4 | 0.447 | 0.477 | 0.479 | 4.215 | 1.642 |
| | 720 | 0.383 | 0.426 | 0.396 | 0.436 | 0.396 | 0.431 | 0.4 | 0.435 | 0.424 | 0.451 | 0.469 | 0.480 | 0.605 | 0.551 | 0.444 | 0.466 | 0.412 | 0.469 | 0.453 | 0.490 | 3.656 | 1.619 |
| | Avg | **0.333** | **0.382** | 0.341 | 0.388 | 0.348 | 0.391 | 0.355 | 0.394 | 0.367 | 0.406 | 0.375 | 0.413 | 0.431 | 0.447 | 0.387 | 0.418 | 0.388 | 0.434 | 0.422 | 0.443 | 3.303 | 1.439 |
| ETTm1 | 96 | 0.308 | 0.348 | 0.281 | 0.340 | 0.299 | 0.354 | 0.288 | 0.345 | 0.305 | 0.361 | 0.306 | 0.358 | 0.299 | 0.343 | 0.311 | 0.366 | 0.326 | 0.390 | 0.510 | 0.492 | 0.626 | 0.560 |
| | 192 | 0.338 | 0.366 | 0.325 | 0.366 | 0.343 | 0.379 | 0.330 | 0.372 | 0.343 | 0.379 | 0.347 | 0.381 | 0.335 | 0.365 | 0.348 | 0.385 | 0.365 | 0.415 | 0.514 | 0.495 | 0.725 | 0.619 |
| | 336 | 0.370 | 0.386 | 0.353 | 0.387 | 0.375 | 0.401 | 0.359 | 0.392 | 0.384 | 0.411 | 0.437 | 0.369 | 0.386 | 0.389 | 0.380 | 0.405 | 0.392 | 0.425 | 0.51 | 0.492 | 1.005 | 0.741 |
| | 720 | 0.426 | 0.416 | 0.410 | 0.413 | 0.431 | 0.439 | 0.406 | 0.421 | 0.439 | 0.431 | 0.466 | 0.425 | 0.424 | 0.422 | 0.443 | 0.444 | 0.446 | 0.458 | 0.527 | 0.493 | 1.133 | 0.845 |
| | Avg | 0.361 | 0.379 | **0.342** | **0.376** | 0.362 | 0.393 | 0.346 | 0.383 | 0.368 | 0.395 | 0.389 | 0.383 | 0.361 | 0.380 | 0.371 | 0.400 | 0.382 | 0.422 | 0.515 | 0.493 | 0.872 | 0.691 |
| ETTm2 | 96 | 0.164 | 0.252 | 0.163 | 0.255 | 0.176 | 0.27 | 0.164 | 0.256 | 0.180 | 0.266 | 0.173 | 0.263 | 0.167 | 0.260 | 0.179 | 0.273 | 0.18 | 0.271 | 0.205 | 0.293 | 0.355 | 0.462 |
| | 192 | 0.218 | 0.290 | 0.217 | 0.290 | 0.232 | 0.304 | 0.223 | 0.296 | 0.234 | 0.302 | 0.228 | 0.301 | 0.224 | 0.303 | 0.242 | 0.315 | 0.252 | 0.318 | 0.278 | 0.336 | 0.595 | 0.586 |
| | 336 | 0.271 | 0.324 | 0.266 | 0.324 | 0.288 | 0.339 | 0.277 | 0.332 | 0.285 | 0.340 | 0.277 | 0.332 | 0.281 | 0.342 | 0.291 | 0.345 | 0.324 | 0.364 | 0.343 | 0.379 | 1.27 | 0.871 |
| | 720 | 0.366 | 0.387 | 0.350 | 0.377 | 0.381 | 0.396 | 0.365 | 0.387 | 0.371 | 0.392 | 0.369 | 0.390 | 0.397 | 0.421 | 0.377 | 0.398 | 0.41 | 0.420 | 0.414 | 0.419 | 3.001 | 1.267 |
| | Avg | 0.255 | 0.313 | **0.249** | **0.311** | 0.269 | 0.327 | 0.257 | 0.318 | 0.267 | 0.325 | 0.262 | 0.322 | 0.267 | 0.332 | 0.272 | 0.333 | 0.292 | 0.343 | 0.310 | 0.357 | 1.305 | 0.797 |
| Weather | 96 | 0.169 | 0.222 | 0.146 | 0.193 | 0.152 | 0.206 | 0.144 | 0.193 | 0.148 | 0.202 | 0.149 | 0.202 | 0.176 | 0.237 | 0.168 | 0.220 | 0.238 | 0.314 | 0.249 | 0.329 | 0.354 | 0.405 |
| | 192 | 0.214 | 0.259 | 0.190 | 0.236 | 0.197 | 0.246 | 0.190 | 0.236 | 0.192 | 0.244 | 0.198 | 0.246 | 0.220 | 0.282 | 0.209 | 0.254 | 0.275 | 0.329 | 0.325 | 0.370 | 0.419 | 0.434 |
| | 336 | 0.260 | 0.294 | 0.242 | 0.278 | 0.246 | 0.285 | 0.244 | 0.280 | 0.243 | 0.282 | 0.245 | 0.286 | 0.265 | 0.319 | 0.266 | 0.295 | 0.339 | 0.377 | 0.351 | 0.391 | 0.583 | 0.543 |
| | 720 | 0.326 | 0.341 | 0.315 | 0.330 | 0.314 | 0.335 | 0.320 | 0.335 | 0.314 | 0.333 | 0.321 | 0.340 | 0.323 | 0.362 | 0.341 | 0.345 | 0.389 | 0.409 | 0.415 | 0.426 | 0.916 | 0.705 |
| | Avg | 0.242 | 0.279 | **0.223** | **0.259** | 0.227 | 0.268 | 0.225 | 0.261 | 0.224 | 0.265 | 0.228 | 0.269 | 0.246 | 0.300 | 0.246 | 0.278 | 0.310 | 0.357 | 0.335 | 0.379 | 0.568 | 0.522 |
| Electricity | 96 | 0.137 | 0.233 | 0.127 | 0.222 | 0.133 | 0.223 | 0.126 | 0.221 | 0.126 | 0.221 | 0.135 | 0.234 | 0.140 | 0.237 | 0.132 | 0.227 | 0.186 | 0.302 | 0.196 | 0.313 | 0.304 | 0.393 |
| | 192 | 0.151 | 0.245 | 0.146 | 0.236 | 0.147 | 0.237 | 0.145 | 0.238 | 0.144 | 0.238 | 0.152 | 0.247 | 0.153 | 0.249 | 0.153 | 0.248 | 0.197 | 0.311 | 0.211 | 0.324 | 0.327 | 0.417 |
| | 336 | 0.167 | 0.261 | 0.161 | 0.256 | 0.166 | 0.265 | 0.164 | 0.256 | 0.160 | 0.255 | 0.169 | 0.268 | 0.169 | 0.267 | 0.168 | 0.264 | 0.213 | 0.328 | 0.214 | 0.327 | 0.333 | 0.422 |
| | 720 | 0.207 | 0.294 | 0.195 | 0.289 | 0.203 | 0.297 | 0.193 | 0.291 | 0.201 | 0.294 | 0.203 | 0.297 | 0.203 | 0.301 | 0.193 | 0.286 | 0.233 | 0.344 | 0.236 | 0.342 | 0.351 | 0.427 |
| | Avg | 0.165 | 0.258 | **0.157** | **0.251** | 0.162 | 0.256 | 0.157 | 0.252 | 0.158 | 0.252 | 0.165 | 0.261 | 0.166 | 0.264 | 0.161 | 0.256 | 0.207 | 0.321 | 0.214 | 0.327 | 0.329 | 0.415 |

particularly prominent at longer horizons (e.g., 336 and 720 steps), highlighting DeCoP's scalability toward robust long-term forecasting.

## H.2 CROSS-DOMAIN FORECASTING RESULTS

We further evaluate the transferability of time series forecasting models under two challenging settings: *in-domain transfer* and *cross-domain transfer*. In the in-domain setting (Table 19), models are trained on one dataset (e.g., ETTh2) and directly evaluated on another from the same domain (e.g., ETTh1). In the cross-domain setting (Table 19), models are pre-trained on one domain (e.g., Weather) and fine-tuned on the target datasets (ETTh1 and ETTm1).

Across both settings, DeCoP$_{MLP}$ demonstrates consistently superior generalization performance. In the in-domain scenario, DeCoP$_{MLP}$ achieves the best or second-best results in 6 out of 8 cases and achieves the lowest average MSE (0.342) and MAE (0.376), outperforming strong baselines like PatchTST (MSE: 0.348, MAE: 0.382) and SimMTM (MSE: 0.351, MAE: 0.383). Similarly, in the more difficult cross-domain setting, DeCoP$_{MLP}$ significantly outperforms all baselines in most transfer paths. For example, on the Weather $\rightarrow$ ETTh1 transfer task, DeCoP$_{MLP}$ achieves an average MSE of 0.411 and MAE of 0.426, clearly surpassing other models, including PatchTST (MSE: 0.426, MAE: 0.448) and SimMTM (MSE: 0.456, MAE: 0.467).

Importantly, DeCoP$_{MLP}$ maintains its leading position even under domain shift, as evidenced by its robust performance on ETTm2 $\rightarrow$ ETTh1 (MSE: 0.412), ETTm1 $\rightarrow$ ETTh1 (MSE: 0.416), and ETTh1 $\rightarrow$ ETTm1 (MSE: 0.346) settings. This indicates strong transferability across temporal structures and seasonal patterns. Furthermore, its linear variant, DeCoP$_{Linear}$, also achieves competitive results with significantly fewer parameters, reinforcing the efficacy of the controllable design for generalizing across domains.

These results highlight the remarkable adaptability of DeCoP in both in-domain and cross-domain scenarios, making it a promising choice for real-world forecasting applications where distribution shifts are common and labeled target-domain data is limited.

Table 19: Complete results of long-term forecasting tasks are presented for the cross-domain setting, where future time points $F \in \{96, 192, 336, 720\}$ are predicted based on the preceding 512 time points. The best results are denoted by **bold**.

| Scenarios | Len | DeCoP$_{Linear}$ | | DeCoP$_{MLP}$ | | PatchTST | | SimMTM | | TF-C | | LaST | | Ti-MAE | | CoST | | TST | | TS2Vec | |
|---|---|---|---|---|---|---|---|---|---|---|---|---|---|---|---|---|---|---|---|---|---|
| | | MSE | MAE | MSE | MAE | MSE | MAE | MSE | MAE | MSE | MAE | MSE | MAE | MSE | MAE | MSE | MAE | MSE | MAE | MSE | MAE |
| ETTh2 ↓ ETTh1 | 96 | 0.362 | 0.391 | 0.365 | 0.394 | 0.380 | 0.411 | 0.372 | 0.402 | 0.596 | 0.569 | 0.362 | 0.42 | 0.399 | 0.424 | 0.376 | 0.362 | 0.401 | 0.425 | 0.436 | 0.43 |
| | 192 | 0.394 | 0.411 | 0.397 | 0.414 | 0.419 | 0.436 | 0.414 | 0.425 | 0.614 | 0.621 | 0.426 | 0.478 | 0.454 | 0.44 | 0.376 | 0.362 | 0.531 | 0.484 | 0.455 | 0.44 |
| | 336 | 0.417 | 0.427 | 0.427 | 0.434 | 0.436 | 0.449 | 0.429 | 0.436 | 0.694 | 0.664 | 0.522 | 0.509 | 0.497 | 0.469 | 0.444 | 0.444 | 0.474 | 0.459 | 0.689 | 0.584 |
| | 720 | 0.438 | 0.459 | 0.446 | 0.462 | 0.457 | 0.474 | 0.446 | 0.458 | 0.635 | 0.683 | 0.46 | 0.478 | 0.515 | 0.492 | 0.517 | 0.51 | 0.471 | 0.469 | 0.489 | 0.49 |
| | Ave | **0.403** | **0.422** | 0.409 | 0.426 | 0.423 | 0.443 | 0.415 | 0.43 | 0.635 | 0.634 | 0.443 | 0.471 | 0.466 | 0.456 | 0.428 | 0.433 | 0.469 | 0.459 | 0.517 | 0.486 |
| ETTm2 ↓ ETTm1 | 96 | 0.306 | 0.347 | 0.283 | 0.342 | 0.294 | 0.35 | 0.297 | 0.348 | 0.61 | 0.577 | 0.304 | 0.388 | 0.333 | 0.378 | 0.32 | 0.364 | 0.327 | 0.364 | 0.422 | 0.434 |
| | 192 | 0.337 | 0.367 | 0.323 | 0.364 | 0.333 | 0.371 | 0.332 | 0.37 | 0.725 | 0.657 | 0.429 | 0.494 | 0.381 | 0.398 | 0.367 | 0.386 | 0.362 | 0.389 | 0.387 | 0.371 |
| | 336 | 0.369 | 0.385 | 0.355 | 0.385 | 0.359 | 0.392 | 0.364 | 0.393 | 0.768 | 0.684 | 0.499 | 0.523 | 0.394 | 0.413 | 0.374 | 0.394 | 0.401 | 0.418 | 0.402 | 0.444 |
| | 720 | 0.424 | 0.416 | 0.406 | 0.414 | 0.407 | 0.414 | 0.41 | 0.431 | 0.927 | 0.759 | 0.422 | 0.45 | 0.455 | 0.453 | 0.479 | 0.503 | 0.437 | 0.437 | 0.481 | 0.432 |
| | Ave | 0.359 | 0.379 | **0.342** | **0.376** | 0.348 | 0.382 | 0.351 | 0.383 | 0.758 | 0.669 | 0.414 | 0.464 | 0.39 | 0.41 | 0.385 | 0.412 | 0.382 | 0.402 | 0.423 | 0.42 |
| ETTm2 ↓ ETTh1 | 96 | 0.363 | 0.392 | 0.366 | 0.394 | 0.385 | 0.411 | 0.388 | 0.421 | 0.968 | 0.738 | 0.428 | 0.454 | 0.433 | 0.431 | 0.403 | 0.426 | 0.389 | 0.413 | 0.483 | 0.48 |
| | 192 | 0.395 | 0.411 | 0.401 | 0.416 | 0.425 | 0.439 | 0.419 | 0.423 | 1.08 | 0.801 | 0.427 | 0.497 | 0.474 | 0.458 | 0.457 | 0.468 | 0.463 | 0.452 | 0.579 | 0.537 |
| | 336 | 0.418 | 0.429 | 0.423 | 0.430 | 0.44 | 0.451 | 0.435 | 0.444 | 1.091 | 0.824 | 0.528 | 0.54 | 0.515 | 0.448 | 0.794 | 0.682 | 0.492 | 0.465 | 0.673 | 0.563 |
| | 720 | 0.441 | 0.461 | 0.459 | 0.465 | 0.482 | 0.488 | 0.468 | 0.474 | 1.226 | 0.893 | 0.527 | 0.537 | 0.496 | 0.488 | 0.739 | 0.617 | 0.468 | 0.468 | 0.729 | 0.62 |
| | Average | **0.404** | **0.423** | 0.412 | 0.426 | 0.433 | 0.447 | 0.428 | 0.441 | 1.091 | 0.814 | 0.503 | 0.507 | 0.464 | 0.456 | 0.598 | 0.548 | 0.453 | 0.45 | 0.616 | 0.55 |
| ETTh2 ↓ ETTm1 | 96 | 0.305 | 0.348 | 0.282 | 0.340 | 0.302 | 0.353 | 0.322 | 0.347 | 0.677 | 0.603 | 0.314 | 0.396 | 0.323 | 0.362 | 0.322 | 0.351 | 0.338 | 0.383 | 0.679 | 0.546 |
| | 192 | 0.341 | 0.370 | 0.323 | 0.365 | 0.342 | 0.375 | 0.332 | 0.375 | 0.718 | 0.638 | 0.587 | 0.545 | 0.37 | 0.395 | 0.331 | 0.373 | 0.394 | 0.408 | 0.673 | 0.551 |
| | 336 | 0.368 | 0.385 | 0.359 | 0.389 | 0.37 | 0.392 | 0.394 | 0.391 | 0.755 | 0.663 | 0.631 | 0.584 | 0.397 | 0.413 | 0.382 | 0.397 | 0.401 | 0.412 | 0.703 | 0.557 |
| | 720 | 0.424 | 0.415 | 0.408 | 0.413 | 0.439 | 0.426 | 0.411 | 0.424 | 0.848 | 0.712 | 0.368 | 0.429 | 0.442 | 0.439 | 0.417 | 0.428 | 0.434 | 0.432 | 0.722 | 0.573 |
| | Average | 0.360 | 0.379 | **0.343** | **0.377** | 0.363 | 0.387 | 0.365 | 0.384 | 0.75 | 0.654 | 0.475 | 0.489 | 0.383 | 0.402 | 0.363 | 0.387 | 0.391 | 0.409 | 0.694 | 0.557 |
| ETTm1 ↓ ETTh1 | 96 | 0.364 | 0.392 | 0.371 | 0.394 | 0.388 | 0.411 | 0.367 | 0.398 | 0.666 | 0.647 | 0.36 | 0.374 | 0.4 | 0.418 | 0.465 | 0.456 | 0.443 | 0.44 | 0.413 | 0.443 |
| | 192 | 0.395 | 0.411 | 0.403 | 0.414 | 0.422 | 0.431 | 0.396 | 0.431 | 0.672 | 0.653 | 0.381 | 0.371 | 0.434 | 0.445 | 0.722 | 0.588 | 0.471 | 0.455 | 0.459 | 0.465 |
| | 336 | 0.419 | 0.430 | 0.428 | 0.431 | 0.449 | 0.449 | 0.471 | 0.437 | 0.626 | 0.711 | 0.472 | 0.531 | 0.51 | 0.467 | 0.712 | 0.586 | 0.462 | 0.455 | 0.614 | 0.554 |
| | 720 | 0.441 | 0.459 | 0.460 | 0.469 | 0.53 | 0.513 | 0.454 | 0.463 | 0.835 | 0.797 | 0.49 | 0.488 | 0.636 | 0.544 | 0.581 | 0.533 | 0.525 | 0.503 | 0.45 | 0.464 |
| | Average | **0.405** | **0.423** | 0.416 | 0.427 | 0.447 | 0.451 | 0.422 | 0.43 | 0.7 | 0.702 | 0.426 | 0.441 | 0.495 | 0.469 | 0.62 | 0.541 | 0.475 | 0.463 | 0.484 | 0.482 |
| ETTh1 ↓ ETTm1 | 96 | 0.309 | 0.350 | 0.283 | 0.340 | 0.293 | 0.344 | 0.29 | 0.348 | 0.672 | 0.6 | 0.295 | 0.387 | 0.311 | 0.355 | 0.308 | 0.355 | 0.315 | 0.354 | 0.681 | 0.545 |
| | 192 | 0.340 | 0.368 | 0.329 | 0.367 | 0.327 | 0.366 | 0.327 | 0.372 | 0.721 | 0.639 | 0.355 | 0.379 | 0.337 | 0.372 | 0.357 | 0.39 | 0.365 | 0.391 | 0.689 | 0.551 |
| | 336 | 0.371 | 0.385 | 0.357 | 0.389 | 0.364 | 0.397 | 0.357 | 0.392 | 0.755 | 0.664 | 0.379 | 0.363 | 0.372 | 0.398 | 0.396 | 0.402 | 0.384 | 0.4 | 0.705 | 0.56 |
| | 720 | 0.422 | 0.414 | 0.414 | 0.417 | 0.409 | 0.417 | 0.409 | 0.423 | 0.837 | 0.705 | 0.403 | 0.431 | 0.422 | 0.433 | 0.419 | 0.423 | 0.428 | 0.426 | 0.722 | 0.571 |
| | Average | 0.361 | 0.379 | **0.346** | **0.379** | 0.348 | 0.381 | 0.346 | 0.384 | 0.746 | 0.652 | 0.353 | 0.39 | 0.36 | 0.39 | 0.37 | 0.393 | 0.373 | 0.393 | 0.699 | 0.557 |
| Weather ↓ ETTh1 | 96 | 0.365 | 0.392 | 0.365 | 0.393 | 0.386 | 0.409 | 0.477 | 0.444 | - | - | - | - | 0.397 | 0.44 | 0.421 | 0.41 | 0.428 | 0.429 | 0.393 | 0.41 |
| | 192 | 0.397 | 0.412 | 0.397 | 0.413 | 0.405 | 0.42 | 0.454 | 0.522 | - | - | - | - | 0.458 | 0.466 | 0.539 | 0.503 | 0.461 | 0.451 | 0.44 | 0.437 |
| | 336 | 0.421 | 0.428 | 0.431 | 0.431 | 0.448 | 0.454 | 0.424 | 0.434 | - | - | - | - | 0.479 | 0.458 | 0.568 | 0.514 | 0.463 | 0.456 | 0.45 | 0.451 |
| | 720 | 0.439 | 0.458 | 0.452 | 0.465 | 0.508 | 0.508 | 0.468 | 0.469 | - | - | - | - | 0.515 | 0.492 | 0.544 | 0.522 | 0.507 | 0.489 | 0.567 | 0.541 |
| | Average | **0.405** | **0.422** | 0.411 | 0.426 | 0.426 | 0.448 | 0.456 | 0.467 | - | - | - | - | 0.462 | 0.464 | 0.518 | 0.487 | 0.465 | 0.456 | 0.463 | 0.46 |
| Weather ↓ ETTm1 | 96 | 0.306 | 0.348 | 0.287 | 0.343 | 0.284 | 0.341 | 0.304 | 0.354 | - | - | - | - | 0.338 | 0.38 | 0.324 | 0.36 | 0.324 | 0.366 | 0.329 | 0.359 |
| | 192 | 0.338 | 0.367 | 0.325 | 0.365 | 0.332 | 0.373 | 0.338 | 0.375 | - | - | - | - | 0.473 | 0.457 | 0.359 | 0.387 | 0.349 | 0.377 | 0.392 | 0.392 |
| | 336 | 0.369 | 0.386 | 0.357 | 0.384 | 0.36 | 0.391 | 0.371 | 0.397 | - | - | - | - | 0.402 | 0.415 | 0.395 | 0.399 | 0.378 | 0.398 | 0.372 | 0.4 |
| | 720 | 0.424 | 0.416 | 0.417 | 0.414 | 0.418 | 0.421 | 0.417 | 0.426 | - | - | - | - | 0.432 | 0.438 | 0.45 | 0.467 | 0.422 | 0.427 | 0.434 | 0.429 |
| | Average | 0.359 | 0.379 | **0.345** | **0.376** | 0.348 | 0.383 | 0.358 | 0.388 | - | - | - | - | 0.411 | 0.423 | 0.382 | 0.403 | 0.368 | 0.392 | 0.382 | 0.395 |

## I FULL BENCHMARK OF TIME SERIES CLASSIFICATION

### I.1 IN- AND CROSS-DOMAIN CLASSIFICATION RESULTS

To evaluate the generalization capability of our method beyond time series forecasting, we further conduct experiments on EEG classification tasks under both in-domain and cross-domain settings (Table 20). In the in-domain setting, models are trained and evaluated on the same dataset (Epilepsy). In cross-domain setting, models are pre-trained on the SleepEEG dataset and fine-tuned on four target datasets: Epilepsy, FD-B, Gesture, and EMG.

In the in-domain scenario, DeCoP achieves state-of-the-art results across all metrics, with an accuracy of 95.53%, F1-score of 92.86%, and the highest average score of 94.20%, outperforming prior methods such as SimMTM (Avg: 92.92%) and TF-C (Avg: 91.03%). Notably, DeCoP achieves strong recall (92.25%) without sacrificing precision (93.51%), indicating its balanced and reliable classification capacity. Under cross-domain transfer, DeCoP continues to exhibit robust generalization. On the SleepEEG → Epilepsy task, DeCoP achieves the highest average score of 94.55%, with competitive performance across all metrics. On SleepEEG → FD-B, DeCoP achieves an average score of 93.97%, significantly outperforming the next-best model (SimMTM, 73.78%). On SleepEEG → EMG, DeCoP highest precision (100%) and F1 value (100%). On SleepEEG → Gesture, DeCoP highest precision (81.67%) and F1 value (80.1%). These results confirm that De-CoP is not only effective in time series forecasting but also excels in classification tasks involving

Table 20: For in-domain setting, we pre-train and fine-tune on the same dataset: Epilepsy. For cross-domain setting, we pre-train the model on SleepEEG and then fine-tune it on different datasets: Epilepsy, FD-B, Gesture and EMG. AVG denotes the average of accuracy and F1 score. The best are denoted by **bold**.

| Scenarios | | Models | Acc (%) | P (%) | R (%) | F1 (%) | Avg (%) |
|---|---|---|---|---|---|---|---|
| In-Domain | Epilepsy ↓ Epilepsy | TS2vec | 92.17 | 93.84 | 81.19 | 85.71 | 88.94 |
| | | CoST | 88.07 | 91.58 | 66.05 | 69.11 | 78.59 |
| | | LaST | 92.11 | 93.12 | 81.47 | 85.74 | 88.93 |
| | | TST | 80.21 | 40.11 | 50.00 | 44.51 | 62.36 |
| | | Ti-MAE | 90.09 | 93.90 | 77.24 | 78.21 | 84.56 |
| | | TF-C | 93.96 | 94.87 | 85.82 | 89.46 | 91.71 |
| | | PatchTST | 89.56 | 90.39 | 89.56 | 80.11 | 84.84 |
| | | SimMTM | 94.75 | 95.60 | 89.93 | 91.41 | 93.08 |
| | | **DeCoP** | 95.53 | 93.51 | 92.25 | 92.86 | **94.20** |
| Cross-Domain | SleepEEG ↓ Epilepsy | TS2vec | 93.95 | 90.59 | 90.39 | 90.45 | 92.20 |
| | | CoST | 88.40 | 88.20 | 72.34 | 76.88 | 82.64 |
| | | LaST | 86.46 | 90.77 | 66.35 | 70.67 | 78.57 |
| | | TST | 80.21 | 40.11 | 50.00 | 44.51 | 63.36 |
| | | Ti-MAE | 89.71 | 72.36 | 67.47 | 68.55 | 79.13 |
| | | TF-C | 94.95 | 94.56 | 80.08 | 91.49 | 93.22 |
| | | PatchTST | 93.27 | 92.51 | 85.57 | 88.48 | 89.96 |
| | | SimMTM | 95.49 | 93.36 | 92.28 | 92.81 | 94.15 |
| | | **DeCoP** | 95.82 | 94.23 | 92.41 | 93.28 | **94.55** |
| | SleepEEG ↓ FD-B | TS2Vec | 47.9 | 43.39 | 48.42 | 43.89 | 45.90 |
| | | CoST | 47.06 | 38.79 | 38.42 | 34.79 | 40.93 |
| | | LaST | 46.67 | 43.9 | 47.71 | 45.17 | 45.92 |
| | | TST | 46.4 | 41.58 | 45.5 | 41.34 | 43.87 |
| | | Ti-MAE | 60.88 | 66.98 | 68.94 | 66.56 | 66.56 |
| | | TF-C | 69.38 | 75.59 | 72.02 | 74.87 | 74.87 |
| | | PatchTST | 80.15 | 82.25 | 85.47 | 83.05 | 86.08 |
| | | SimMTM | 69.4 | 74.18 | 76.41 | 75.11 | 72.26 |
| | | **DeCoP** | 93.04 | 94.92 | 94.90 | 94.90 | **93.97** |
| | SleepEEG ↓ Gesture | TS2Vec | 69.17 | 65.45 | 68.54 | 65.70 | 67.44 |
| | | CoST | 68.33 | 65.3 | 68.33 | 66.42 | 67.38 |
| | | LaST | 64.17 | 70.36 | 64.17 | 58.76 | 61.47 |
| | | TST | 69.17 | 66.6 | 69.17 | 66.01 | 67.59 |
| | | Ti-MAE | 71.88 | 70.35 | 76.75 | 68.37 | 70.13 |
| | | TF-C | 76.42 | 77.31 | 74.29 | 75.72 | 76.07 |
| | | PatchTST | 74.17 | 72.18 | 74.17 | 71.40 | 72.78 |
| | | SimMTM | 80.00 | 79.03 | 80.00 | 78.67 | 79.34 |
| | | **DeCoP** | 81.67 | 80.99 | 81.67 | 80.10 | **80.89** |
| | SleepEEG ↓ EMG | TS2Vec | 78.54 | 80.4 | 67.85 | 67.66 | 73.10 |
| | | CoST | 53.65 | 49.07 | 42.1 | 35.27 | 44.46 |
| | | LaST | 66.34 | 79.34 | 63.33 | 72.55 | 69.45 |
| | | TST | 78.34 | 77.11 | 80.3 | 68.89 | 73.62 |
| | | Ti-MAE | 69.99 | 70.25 | 63.44 | 70.89 | 70.44 |
| | | TF-C | 81.71 | 72.65 | 81.59 | 76.83 | 79.27 |
| | | PatchTST | 90.24 | 82.96 | 82.95 | 82.91 | 82.94 |
| | | SimMTM | 97.56 | 98.33 | 98.04 | 98.14 | 97.85 |
| | | **DeCoP** | 100.00 | 100.00 | 100.00 | 100.00 | **100.00** |

physiological signals. Its consistent performance across datasets and domains highlights its strong inductive bias and adaptability, especially in low-resource transfer scenarios. Unlike prior models that exhibit strong performance on only specific metrics (e.g., high recall but low precision), DeCoP demonstrates balanced, high-quality predictions across all evaluation dimensions.