# OpenReview forum: "DeCoP: Enhancing Self-Supervised Time Series Representation with Dependency Controlled Pre-training"
_ICLR.cc/2026/Conference — ICLR 2026 Conference Withdrawn Submission_

### Official Review · Reviewer_HKk6 · 2025-10-23

**Soundness:** 2
**Presentation:** 2
**Contribution:** 2
**Rating:** 2
**Confidence:** 5

**Summary:**

The paper proposes DeCoP, a self-supervised pre-training framework for time-series that combines: (1) IPN to blend instance- and patch-level statistics for robustness to distribution shifts while retaining local semantics; (2) DCL to control receptive fields across multiple temporal scales; (3) ICM to build time-invariant positives via frequency-domain masking and a contrastive loss. Across 6 forecasting and 4 classification datasets, DeCoP reports better accuracy with fewer FLOPs, strong cross-domain transfer, and improved classification.

**Strengths:**

DeCoP effectively integrates IPN, DCL, and ICM into a unified framework, combining multi-scale dependency modeling with frequency-domain contrastive learning to enhance generalization and stability,  achieving strong performance with low computational cost.

**Weaknesses:**

1. The frequency-domain masking mechanism in ICM raises potential concerns about cross-sample information leakage, since the frequency selection is based on batch- or channel-level averaged amplitudes. This may implicitly introduce dependencies between samples, particularly under cross-domain or distribution-shift scenarios. A clearer explanation or ablation demonstrating that such leakage does not affect training fairness would be valuable.

2. The stability and interpretability of the learnable coefficient α in IPN are insufficiently analyzed. Although Figure 6 shows qualitative distribution alignment, there is no quantitative study or sensitivity test on how α evolves across datasets or initialization schemes. Such analysis would clarify whether the IPN layer remains stable under highly non-stationary data.

3. Reporting averages over multiple random seeds and performing significance testing would make the empirical claims more convincing.

4. Comparison set of baselines remains somewhat limited. Including comparisons with more recent multi-scale or frequency-aware architectures would help position the work more clearly within the state of the art.

5. While FLOPs are reported, it is unclear whether FFT/iFFT operations and contrastive heads are included. Providing a more detailed breakdown of computational and memory cost between pre-training and fine-tuning phases would highlight the real efficiency advantage.

**Questions:**

Please see the Weaknesses, and the following:

1. The paper states that the Instance-level Contrastive Module (ICM) selects “time-invariant” frequencies using averaged amplitudes across the batch or channel (Eqs. 12–13). Could the authors clarify the exact scope of this averaging? Is it computed only within each batch of the source-domain data, or does it aggregate statistics from multiple domains or entire datasets? In cross-domain fine-tuning, is the same set of frequencies frozen, re-estimated per domain, or adaptively updated? It would help to describe how the model avoids potential cross-sample leakage during frequency selection.

2. How sensitive is the ICM to the hyperparameters 𝐾 and 𝑀 used for selecting top-K or top-M frequencies? An analysis of performance stability across different masking settings would clarify the robustness of the ICM component.

3.  The learnable mixing factor between instance-level and patch-level statistics (Eq.~7) is central to the IPN design. Could the authors provide more details on how $\alpha$ evolves during training and across datasets? Is $\alpha$ constrained within $[0,1]$ via sigmoid activation, and is there any regularization applied? Visualizing or quantifying $\alpha$’s dynamics might help demonstrate whether IPN consistently balances local and global normalization, especially under severe distribution shifts.

4. In Table 5, DeCoP demonstrates robustness when transferring from ETTh2 → ETTh1 under limited labels. Are all hyperparameters (e.g., masking ratio, learning rate, optimizer) kept identical between pretraining and fine-tuning? If not, what are the adjusted values? Providing a clear protocol for transfer experiments would strengthen the reproducibility of these results.

5. While DeCoP is compared against strong baselines such as PatchTST and SimMTM, have the authors evaluated it against newer hybrid or frequency-enhanced architectures (e.g., TimeMixer$^{[1]}$, FreTS$^{[2]}$, DiffTime$^{[3]}$, and Affirm$^{[4]}$)?

[1] Timemixer: Decomposable multiscale mixing for time series forecasting, ICLR 2025.

[2] Frequency-domain MLPs are more effective learners in time series forecasting, Nips 2023.

[3] On the constrained time-series generation problem, Nips 2023.

[4] Affirm: Interactive mamba with adaptive fourier filters for long-term time series forecasting, AAAI 2025.

---

### Official Review · Reviewer_pq5T · 2025-10-30

**Soundness:** 2
**Presentation:** 3
**Contribution:** 2
**Rating:** 4
**Confidence:** 4

**Summary:**

This paper proposes DeCoP (Dependency Controlled Pre-training), a self-supervised pre-training framework for time series. The core idea is to explicitly model dynamic, multi-scale temporal dependencies that evolve under distribution shifts.The framework consists of three key modules: Instance-wise Patch Normalization (IPN): alleviates distribution shifts at the input level while preserving local semantic features; Hierarchical Dependency Controlled Learning (DCL): adaptively captures short- and long-term dependencies through multi-scale temporal windows;Instance-level Contrastive Module (ICM): leverages frequency-domain filtering to generate time-invariant positive pairs, thereby enhancing global generalization. Experimental results show that DeCoP achieves superior performance on both time series forecasting and classification datasets.

**Strengths:**

The paper clearly identifies dynamic dependency modeling and distribution shift as key bottlenecks in time-series SSL. The proposed three-module design forms a coherent framework that aligns well with these goals.

DeCoP achieves consistent improvements over strong baselines such as PatchTST and SimMTM, with lower FLOPs, demonstrating that the proposed mechanisms contribute to efficiency and generalization.

The paper includes both forecasting and classification tasks, as well as cross-domain transfer evaluations, which help validate the robustness of the approach.

The method is clearly described and the figures help illustrate the intuition behind dependency-controlled learning.

**Weaknesses:**

DCL’s hierarchical windowing resembles multi-scale Transformers such as Autoformer, FEDformer, or TimeMixer.
ICM’s frequency-domain contrastive design is closely related to FreRA and TF-C.
Overall, the contribution is integrative rather than fundamentally novel.

Although IPN is proposed as a novel normalization strategy to handle distribution shifts, the paper does not compare it with alternatives such as IN-flow, FAN or RevIN.
Without such comparison, the claimed advantages of IPN in stability and performance remain unverified.

Despite claiming open-sourcing, the code repository is currently empty or inaccessible, raising concerns about reproducibility and limiting the community’s ability to validate the approach.

The paper claims that the model enables explicit dependency control, yet lacks qualitative or visual analysis to demonstrate how dependencies evolve or how frequency-domain filtering affects the learned representations.

The hierarchical DCL module and patch-wise IPN normalization could introduce computational overhead for long or high-frequency sequences. Although efficiency is mentioned, the paper provides no concrete runtime comparison or complexity analysis to support this claim.

**Questions:**

See weakness.

---

### Official Review · Reviewer_3VNA · 2025-11-01

**Soundness:** 2
**Presentation:** 3
**Contribution:** 2
**Rating:** 4
**Confidence:** 3

**Summary:**

DeCoP is a specially designed self-supervised pre-training framework for time series data. It focuses on solving two major challenges: changing temporal dependencies caused by distribution shifts, and the need to capture both short-term and long-term patterns. By introducing Instance-wise Patch Normalization (IPN), Dependency Controlled Learning (DCL), and an Instance-level Contrastive Module (ICM), DeCoP effectively models dynamic, multi-scale dependencies. The results show that DeCoP outperforms other state-of-the-art methods like PatchTST and SimMTM, achieving better performance with lower computational cost.

**Strengths:**

1. This paper focuses on an important challenge in self-supervised learning for time series data, i.e., how to effectively model dynamic temporal dependencies under distribution shifts.
2. DeCoP achieves state-of-the-art performance in both forecasting and classification tasks while maintaining outstanding computational efficiency.

**Weaknesses:**

1. I would like to further assess the role of $\alpha$  in IPN. Please evaluate the impact of freezing or randomizing α compared to learning it. Additionally, how does the learned $\alpha$ vary across different datasets? What is the effect of designing $\alpha$ with finer granularity, such as channel-level or patch-level?  If  $\alpha$ is globally shared, how does the model ensure effective balancing of local and global statistics across diverse data distributions?
2. Is normalization and denormalization in IPN stable during fine-tuning? Specifically, what happens when patch variance is close to zero? Are there safeguards to ensure numerical stability in such cases?

3. The DCL module relies on a set of predefined window sizes (W_k) for hierarchical dependency modeling. Why not use an adaptive mechanism (e.g., mixing layers in TimesNet or the adaptive spectral block in TSLANet) to potentially outperform fixed-size windows in capturing dynamic dependencies? Additionally, is there any prior or heuristic for selecting appropriate (W_k) values across different datasets?
4. The paper seems to introduce many parameters but does not fully discuss their sensitivity. For example, the ICM module introduces top-K and top-M frequency filtering as a key component of positive pair generation. However, the paper does not provide an in-depth analysis of the model’s sensitivity to the choice of β, which governs the threshold for selecting global versus local frequency components.

**Questions:**

See the weaknesses for details.

---

### Official Review · Reviewer_WZD8 · 2025-11-01

**Soundness:** 3
**Presentation:** 3
**Contribution:** 2
**Rating:** 4
**Confidence:** 4

**Summary:**

The paper introduces DeCoP, a framework designed to enhance self-supervised representation learning for time series. It explicitly models dynamic and multi-scale temporal dependencies that evolve due to distribution shifts. At the input level, it introduces Instance-wise Patch Normalization to balance local and global statistics, mitigating distribution shifts while preserving fine-grained semantics. At the latent level, Dependency Controlled Learning adaptively captures short- and long-term dependencies, while an Instance-level Contrastive Module improves global generalization via time-invariant positive pairs. DeCoP achieves state-of-the-art performance on ten datasets with reduced computational cost, improving both forecasting and classification tasks.

**Strengths:**

1. The paper proposes a well-motivated framework that explicitly tackles distribution shifts and dynamic dependencies, a core challenge in time series pre-training.


2. Experimental results are comprehensive, covering both forecasting and classification across in-domain and cross-domain settings, with clear efficiency gains over strong baselines.

**Weaknesses:**

1. Some design motivations lack theoretical or empirical justification, particularly regarding patch normalization and dependency control under varying statistics.


2. The novelty is partially incremental, as several ideas,e.g., multi-scale learning, contrastive consistency, overlap with prior self-supervised frameworks; clearer differentiation is needed.

**Questions:**

1. In line 57 of the Introduction, the discussion mentions MTM-type methods, but it overlooks multi-scale approaches in contrastive learning (CL) methods. Could the authors clarify or include related CL-based multi-scale methods for a more complete comparison?

2. If patch statistics differ significantly due to distribution shifts, anomalies, or patch size choice, the model may struggle to learn stable global features, and the claimed adaptive benefit of combining local and global statistics lacks quantitative support. Could the authors provide quantitative evidence to support the claim of improved stability?

3. The operations of splitting into patches and defining a set of window sizes may be somewhat redundant, potentially leading to a disjoint approach in addressing distribution shifts versus multi-scale patterns. Can the authors clarify the necessity and interaction of these two operations?

4. In the Related Work section, the two subsections appear to have an inclusion relationship. Could the authors clarify the distinction and avoid potential redundancy?

---

### Note · Authors · 2025-11-12

I have read and agree with the venue's withdrawal policy on behalf of myself and my co-authors.